# Genomic analyses reveal the stepwise domestication and genetic mechanism of curd biogenesis in cauliflower

**Rui Chen** [1,8] ✉, **Ke Chen** [2,3,8], **Xingwei Yao** [1,8], **Xiaoli Zhang** [1,8], **Yingxia Yang** [1], **Xiao Su** [2], **Mingjie Lyu** [1], **Qian Wang** [1], **Guan Zhang** [1], **Mengmeng Wang** [1], **Yanhao Li** [1], **Lijin Duan** [1], **Tianyu Xie** [1], **Haichao Li** [1,4], **Yuyao Yang** [1,4], **Hong Zhang** [1,4], **Yutong Guo** [1,4], **Guiying Jia** [1,4], **Xianhong Ge** [5], **Panagiotis F. Sarris** [6,7], **Tao Lin** [2] ✉ & **Deling Sun** [1] ✉

Cauliflower (*Brassica oleracea* L. var. *botrytis*) is a distinctive vegetable that supplies a nutrient-rich edible inflorescence meristem for the human diet. However, the genomic bases of its selective breeding have not been studied extensively. Herein, we present a high-quality reference genome assembly C-8 (V2) and a comprehensive genomic variation map consisting of 971 diverse accessions of cauliflower and its relatives. Genomic selection analysis and deep-mined divergences were used to explore a stepwise domestication process for cauliflower that initially evolved from broccoli (Curd-emergence and Curd-improvement), revealing that three MADS-box genes, *CAULIFLOWER1* (*CAL1*), *CAL2* and *FRUITFULL* (*FUL2*), could have essential roles during curd formation. Genome-wide association studies identified nine loci significantly associated with morphological and biological characters and demonstrated that a zinc-finger protein (BOB06G 135460) positively regulates stem height in cauliflower. This study offers valuable genomic resources for better understanding the genetic bases of curd biogenesis and florescent development in crops.

*Brassica oleracea*, the CC-genome diploid in the Triangle of U[1], is characterized by its remarkable morphological diversity, bearing specialized leafy, stem or floral organs represented by Chinese kale (*B. oleracea* var. *alboglabra*), kohlrabi (var. *gongylodes*), Brussels sprouts (var. *gemmifera*), cabbage (var. *capitata*), broccoli (var. *italica*) and cauliflower (var. *botrytis*). However, the great variety of wild species, intraspecific crossability[2] and strong self-incompatibility[3] raise serious

challenges for investigating the domestication history of *B. oleracea*, the authentic relationships among its subspecies and its bona fide ancestral source. Recently, multiple pieces of evidence have indicated that the Aegean-endemic *Brassica cretica* might be the closest wild relative of the currently cultivated *B. oleracea*[4].

Among the *B. oleracea* subspecies, cauliflower is an economically important vegetable crop possessing unique flavor, high nutritional

[1]State Key Laboratory of Vegetable Biobreeding, Tianjin Academy of Agricultural Sciences, Tianjin, China. [2]Beijing Key Laboratory of Growth and Developmental Regulation for Protected Vegetable Crops, College of Horticulture, China Agricultural University, Beijing, China. [3]Key Laboratory of Weed Control in Southern Farmland, Ministry of Agriculture and Rural Affairs, Hunan Academy of Agricultural Sciences, Changsha, China. [4]College of Life Sciences, Nankai University, Tianjin, China. [5]National Key Laboratory of Crop Genetic Improvement, College of Plant Science and Technology, Huazhong Agricultural University, Wuhan, China. [6]Institute of Molecular Biology and Biotechnology, Foundation for Research and Technology-Hellas, Heraklion, Greece. [7]Department of Biology, University of Crete, Heraklion, Greece. [8]These authors contributed equally: Rui Chen, Ke Chen, Xingwei Yao, Xiaoli Zhang. ✉e-mail: chenrui.taas@gmail.com; lintao35@cau.edu.cn; sundeling1961@163.com

value and anticancer activity[2]. The global production of cauliflower and broccoli is continuously increasing and reached over 25.5 million tons with a net value of 14.1 billion US dollars in 2020 (http://faostat.fao.org/). Cauliflower and broccoli, regarded as the 'arrested inflorescence' lineage, are speculated to have been domesticated ~2,500 years ago[5]. Cultivated cauliflower is generally divided into loose-curd and compact-curd classes according to its curd solidity[6], although the detailed population structure of cauliflower has not been well clarified owing to its short evolutionary history and narrow genetic background[7,8]. Until recently, three ecotypes with different maturity levels had been roughly determined in cauliflower, excluding Romanesco cauliflower[5]. Now, two draft genome sequences of cauliflower have been reported[9,10]. These have expanded our understanding of modern cauliflower demography and the phenotypic variation that has occurred during differentiation and domestication. However, owing to the low contiguity of the genome sequence, the lack of high-density markers, and the limited sampling of cauliflower and ancestral wild accessions in previous studies[4,5,10,11], the genome-wide effects of selection and the genetic mechanisms underlying important agronomic traits in cauliflower remain poorly understood.

Curd biogenesis is a complex process regulated by multiple developmental signals and environmental factors[10,12,13], involving vernalization[14], photoperiod[15], gibberellin[16], and autonomous[17] flowering-related pathways. In cauliflower and *Arabidopsis*, several important curd-biogenesis-related genes have been identified, including MADS-box genes *CAULIFLOWER* (*CAL/AGL10*), *APETALA1* (*AP1/AGL7*)[18], *FRUITFULL* (*FUL/AGL8*)[19], *SUPPRESSOR OF OVEREXPRESSION OF CO 1* (*SOC1/AGL20*)[20], *AGAMOUSLIKE 24* (*AGL24*)[21] and *XAANTAL2* (*XAL2/AGL14*)[22], as well as phosphatidylethanolamine-binding protein *TERMINAL FLOWER 1* (*TFL1*)[23] and a plant-specific transcription factor gene, *LEAFY* (*LFY*)[24]. The nested-spiral pattern of cauliflower curd has been preliminarily deciphered using a three-dimensional computational model[13]. However, our knowledge is still segmental and the underlying genetic mechanisms remain elusive.

In this study, we have updated the high-quality reference genome assembly of cauliflower C-8 (V2) and present a comprehensive genomic variation map derived from the resequencing of 971 diverse cauliflower accessions and their relatives. Using these data, we performed population genomic analyses to genetically dissect the evolutionary relationships among *B. oleracea* subspecies and explored the molecular mechanism of curd biogenesis and seven important agronomic traits. Further functional experiments demonstrated that a zinc-finger protein (BOB06G135460) positively regulates SH and three significantly associated biomass traits in cauliflower. This work provides information to better understand the nature of cauliflower and lays a solid foundation for future germplasm utilization and improvements in cauliflower breeding.

## Results

### De novo assembly and annotation of the cauliflower genome

We updated a highly contiguous and complete genome sequence of the cauliflower inbred line C-8 (V2) using an integrated approach including PacBio SMRT sequencing, Bionano optical mapping and Hi-C technologies, supplemented with Illumina whole-genome shotgun data[9] (Supplementary Fig. 1a). As a result, we achieved a high-quality assembly comprising 557 contigs with a contig N50 of 10.57 Mb and a total genome size of 568.52 Mb that anchors and orients 557.11 Mb (approximately 98%) onto nine pseudochromosomes. Compared with the previously published C-8 genome[9], this updated version is markedly improved with better completeness and contiguity. Moreover, the C-8 (V2) genome exhibits greater advantages in terms of contig N50 and genome quality (higher BUSCO value and lower gap numbers) compared with the recently released cauliflower genome 'Korso'[10] (Supplementary Fig. 2 and Table 1).

By integrating evidence from ab initio predictions, RNA sequencing (RNA-seq) data and homology searching, a total of 57,983 protein-coding genes were functionally annotated. Approximately 331.36 Mb (58.30%) of the updated genome was identified as consisting of repeat sequences. Of these, *Gypsy*-type (13.42%) and *Copia*-type (10.14%) long terminal repeats were the predominant repetitive elements in the entire genome (Supplementary Table 2). In addition, nine potential centromeric regions were distinguished across the entire genome, ranging from 1.9 to 6.9 Mb (Supplementary Fig. 1b). These findings demonstrate the high quality and coverage of the C-8 (V2) genome sequence and indicate that it provides an ideal model system for studying curd organ development and a preferred resource for cauliflower breeding.

### Genomic variation of cauliflower and its relatives

To achieve a comprehensive genomic variation map, we collected a total of 820 diverse cauliflower and *B. oleracea* accessions for whole-genome resequencing and downloaded 151 additional accessions derived from up-to-date data available from previous studies[11,25]. In total, we acquired 726 cauliflower accessions representing broad genetic and phenotypic diversity, as well as 43 accessions for broccoli, 50 for cabbage, 13 for Brussels sprouts, 28 for kohlrabi, 59 for Chinese kale, and 30 for wild relatives and other *B. oleracea* subspecies (Supplementary Table 3). Resequencing of these accessions yielded 7.59 Tb of sequencing data, with an average depth of 7.8× and coverage of 90.55% of the C-8 (V2) genome. After alignment with the cauliflower reference genome, we detected a final set of 17,917,317 single-nucleotide polymorphisms (SNPs) and 10,831,040 insertions and deletions (InDels), much more than previously reported for *B. oleracea*[11]. Among these variants, 1,872,979 (3.11%) nonsynonymous SNPs and 720,309 (1.47%) frameshift InDels were located within coding regions of 55,927 (96.45%) annotated genes. In addition, 903,486 variants in 53,491 (92.25%) genes showed potentially large effects, leading to truncated or elongated transcripts, frameshift mutations or other disruptions of protein-coding capacity (Supplementary Tables 4 and 5). These variants provide a valuable resource for functional genomic researches and marker-assisted breeding in *B. oleracea*.

### Evolutionary relationships among *B. oleracea* subspecies

Although the representative *B. oleracea* subspecies are easily distinguished based on their specialized edible organs, the exact evolutionary relationships among *B. oleracea* subspecies remain uncertain because of their frequent genetic exchanges[4,10,11]. To explore the phylogenetic relationships among these plants, we used a subset of 69,275 SNPs at fourfold degenerate sites (4d-SNPs) among 971 *B. oleracea* accessions to build a maximum-likelihood (ML) tree. Evidence from the ML tree, model-based clustering and principal component analysis (PCA) supported four major clades: clade 1, solely composed of Chinese kale; clade 2, mainly including kohlrabi, Brussels sprouts and cabbage; and clade 3 and clade 4, corresponding to broccoli and cauliflower, respectively (Fig. 1a,b and Supplementary Fig. 3). These results are mostly in agreement with those of previous studies[4,5,10,11], but they are more informative with respect to the identity of *B. cretica* and Chinese kale, as well as the classification of *B. oleracea* subspecies.

Clade 1 was closest to the phylogenetic root and occupied a distinct position in the PCA results (Fig. 1a,c). Clade 1 had a relatively lower level of nucleotide diversity ($\pi_{clade1} = 1.08 \times 10^{-3}$) than clade 2 ($\pi_{clade2} = 1.29 \times 10^{-3}$), implying that infrequent genetic exchange occurred, perhaps owing to its early geographic isolation (Fig. 1d). This is consistent with the historical record in which clade 1 was introduced to China from Europe during the Northern and Southern Dynasties ~AD 420–589[26] and evolved as an independent population. Our analysis assigned kohlrabi, lacinato kale, curly kale, Brussels sprouts, savoy cabbage, kale and cabbage into clade 2. We found that these subspecies shared closer relationships among the *B. oleracea* subspecies, suggesting that they may have undergone widespread gene exchange during their differentiation (Fig. 1a,b). Notably, eight wild accessions within clade 2 might be feral plants derived from intraspecific hybridization

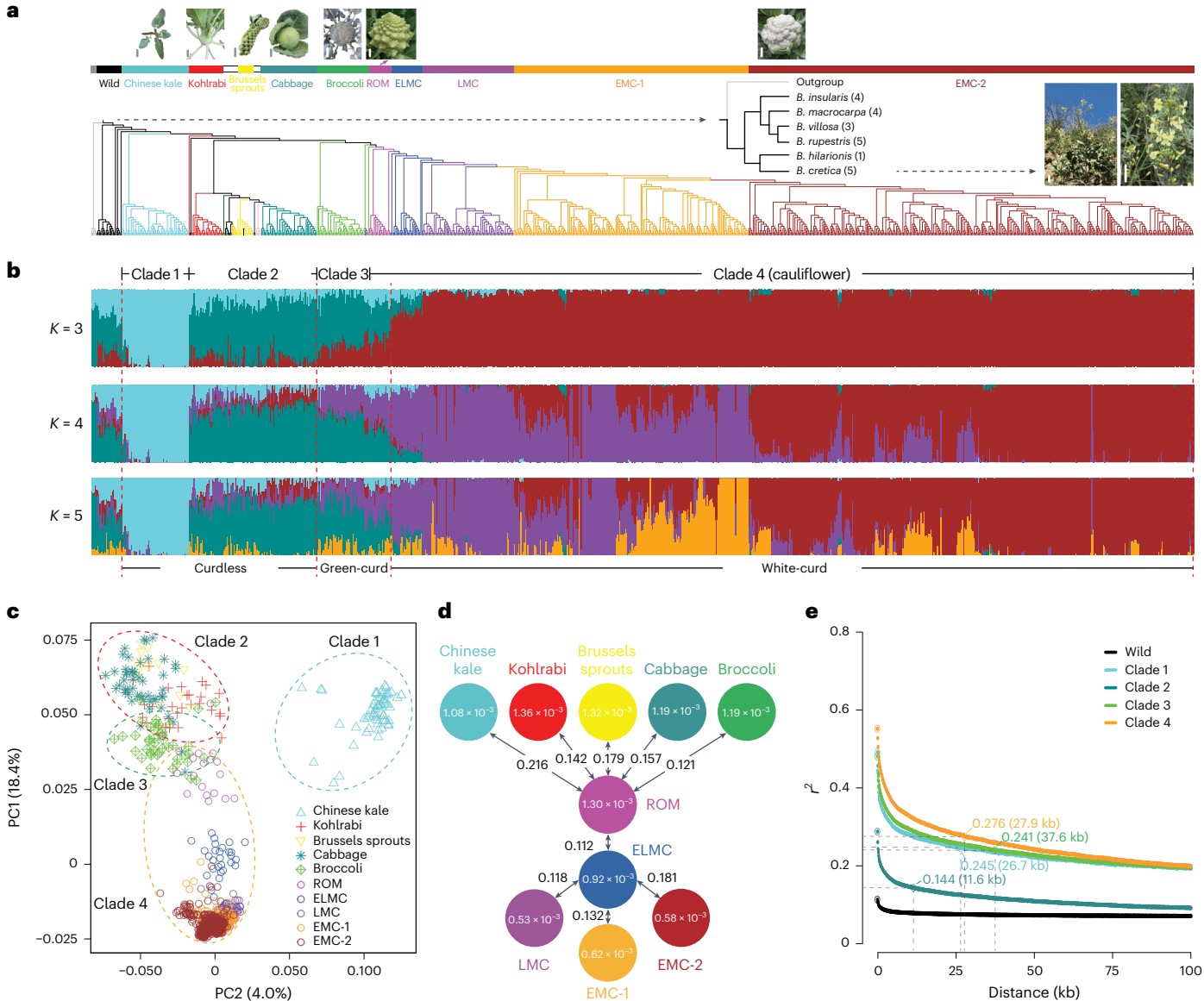

**Fig. 1 | Genomic relationships of 971 *B. oleracea* accessions. a**, Clades and groups including wild and major subspecies of *B. oleracea*, along with their relationships, illustrated using a phylogenetic tree. Different colors represent different groups as follows: gray, outgroup (*B. rapa* and *B. nigra*); black, wild and feral-type; cyan, Chinese kale; red, kohlrabi; dark green, lacinato kale; gold, curly kale; yellow, Brussels sprouts; dark blue, savoy cabbage; pink, kale; dark cyan, cabbage; green, broccoli; dark purple, purple cauliflower; magenta, ROM; blue, ELMC, purple, LMC; orange, EMC-1; brown, EMC-2. White and gray bars indicate 5 cm. **b**, Results of model-based clustering when $K = 3$, 4 and 5. Red dashed lines indicate three categories as labeled: Curdless, Green-curd and White-curd. **c**, PCA analysis based on 1,564 4d-SNPs. **d**, Summary of nucleotide diversity ($\pi$) and population divergence ($F_{ST}$) among major *B. oleracea* subspecies and each cauliflower group. **e**, LD decay. Dashed lines and colored dots indicate the half-maximum distance and corresponding $r^2$ values, respectively.

or escape from domestication[4,27]. Compared with the 22 wild relatives at the base of the phylogenetic tree ($\pi_{wild} = 2.31 \times 10^{-3}$), these feral accessions had lower nucleotide diversity ($\pi_{feral} = 1.52 \times 10^{-3}$), but higher than that of the entire clade 2 ($\pi_{clade2} = 1.29 \times 10^{-3}$), indicating a substantial genetic difference between feral and authentic wild relatives (Fig. 1d and Supplementary Table 6). This speculation was also supported by the values of the inbreeding coefficient, which differed markedly between the feral and wild accessions (Supplementary Fig. 4).

The phylogenetic tree and model-based analysis showed that the floral-organ-specialized clade 4 probably directly evolved from clade 3 rather than from wild relatives, consistent with previous speculations[28,29]. Compared with the clade 1, clade 2 and clade 3 accessions, the clade 4 accessions showed the lowest nucleotide diversity ($\pi_{Clade4} = 0.73 \times 10^{-3}$) and an intimate relationship with clade 3 ($F_{ST} = 0.186$) (Supplementary Fig. 5 and Table 6). The linkage

disequilibrium (LD) decay indicated that clade 4 had moderate physical distance between SNPs (27.9 kb) compared with the other clades, in which it ranged from 11.6 to 37.6 kb (Fig. 1e). Notably, when $K = 3$, model-based clustering showed a clear and gradual tendency from clade 3 (hereafter 'broccoli') to clade 4 (hereafter 'cauliflower'), suggesting the evolutionary pathway of cauliflower (Fig. 1b).

## Stepwise domestication of cauliflower

Cauliflower has undergone a short evolutionary history (~2,500 years), and strong bottlenecks may have occurred during its domestication[5]. To date, the population structure of cauliflower has remained unclear. Based on the phylogenetic tree, plant architecture and maturity levels, we assigned the 726 cauliflower accessions into five groups: ROM (Romanesco cauliflower), ELMC (extremely late-maturing cultivars), LMC (late-maturing cultivars), EMC-1 (early-maturing cultivars) and EMC-2.

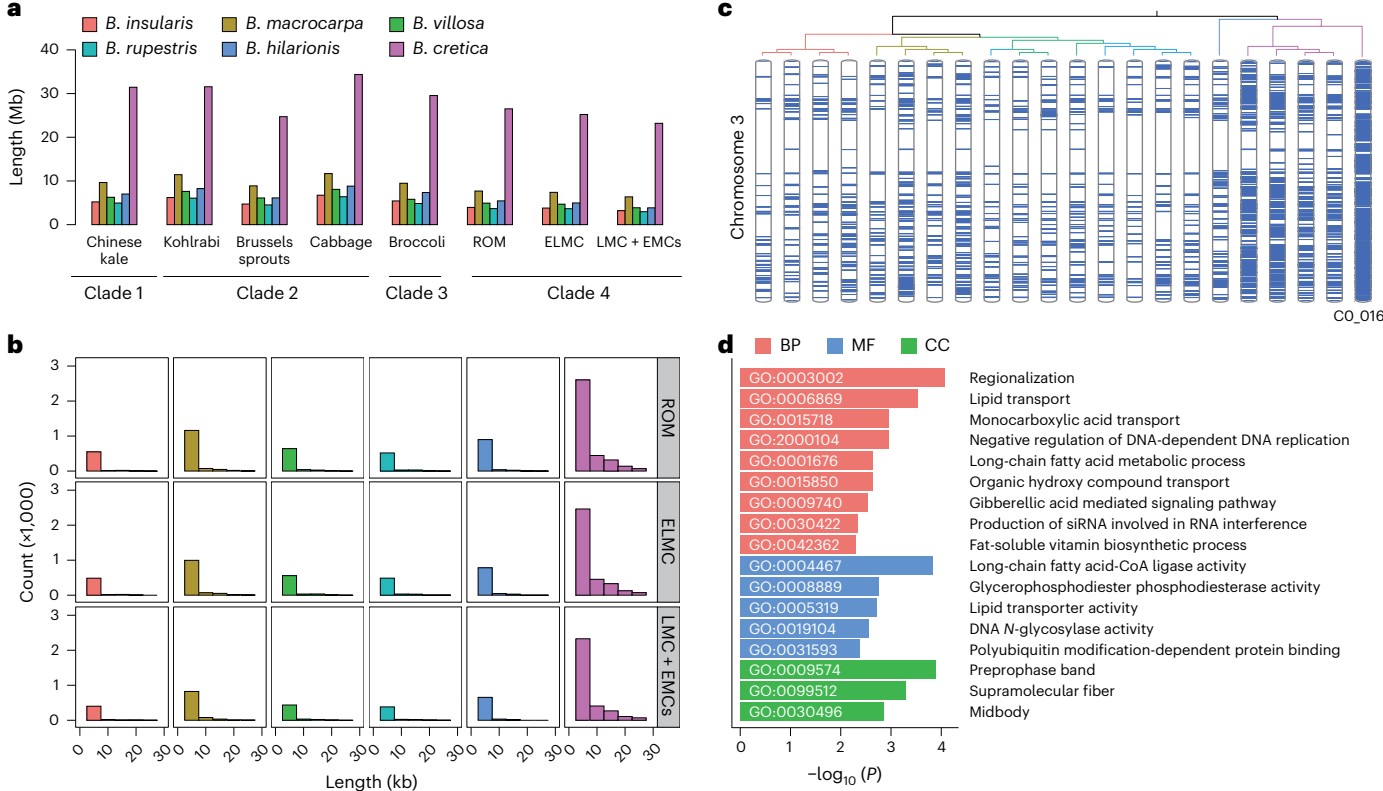

**Fig. 2 | Putative admixture and ancestral inference. a**, Average length of identical windows between wild accessions and *B. oleracea* subspecies. **b**, Histograms of identical fragments. A single representative accession was selected for each wild species as follows: SRR6453800, *B. insularis*; C0_0166, *B. macrocarpa*; SRR6453822, *B. villosa*; SRR6453618, *B. rupestris*; SRR6453871,

*B. hilarionis*; SRR9331105, *B. cretica*. **c**, Diagram of inferred syntenic regions between the LEE groups of cauliflower and 22 wild accessions in chromosome 3. **d**, GO enrichment analysis of genes from identical regions between *B. cretica* C0_0162 and the LEE groups. *P* values were adjusted using the Benjamini–Hochberg false discovery rate correction.

The ROM group was the nearest phylogenetic neighbor and showed the lowest level of genetic differentiation ($F_{ST} = 0.121$) from the broccoli accessions (Fig. 1a,d). Among these groups, the ROM group bears light green and pyramidal shaped curds, making its appearance different from that of the other cauliflower groups. Moreover, ROM displayed the highest level of nucleotide diversity ($\pi_{ROM} = 1.30 \times 10^{-3}$) among the five cauliflower groups, suggesting that it may be the predecessor of cauliflower cultivars and may have had a transitional role during cauliflower differentiation.

The ELMC accessions are regarded as valuable germplasms for cauliflower breeding, owing to their excellent properties of cold hardiness and disease resistance. Among the five cauliflower groups, the lowest $F_{ST}$ value was that between the ROM and ELMC groups (0.112), followed by that between the ELMC and LMC groups (0.118) (Fig. 1d). In addition, genetic diversity decreased from the ROM ($\pi_{ROM} = 1.30 \times 10^{-3}$) to the ELMC ($\pi_{ELMC} = 0.92 \times 10^{-3}$) group, and then to the LMC and EMC groups ($\pi_{average(LMC+EMCs)} = 0.58 \times 10^{-3}$) (Fig. 1d and Supplementary Table 6). The PCA plots also supported the transitional roles of the ROM and ELMC groups, which occupied bridging positions between broccoli and the majority of the cauliflower groups (LMC, EMC-1 and EMC-2) (Fig. 1c and Supplementary Fig. 3). Notably, the LMC, EMC-1 and EMC-2 groups (hereafter 'LEE' groups) were tightly clustered in the PCA results (Fig. 1c), suggesting their highly similar genetic backgrounds and the strong bottlenecks that cauliflower experienced. Taken together, these results indicate that cauliflower has undergone a one-way and stepwise domestication route that yielded the ROM and ELMC groups from broccoli and further improved into the early-maturity cauliflower cultivars.

## Genomic evidence for the wild ancestor of *B. oleracea*

The 'C9' wild relatives of *B. oleracea* contain nine chromosome pairs and are generally considered to be the ancestral origin. They are mainly located in the Mediterranean region and are able to produce fertile hybrids through crossing with *B. oleracea* subspecies[30,31]. To identify the authentic progenitor of cauliflower and *B. oleracea*, we inferred potential identical genomic regions by comparing each *B. oleracea* subspecies with 22 'C9' wild relatives (*Brassica insularis*, *Brassica macrocarpa*, *Brassica villosa*, *Brassica rupestris*, *Brassica hilarionis* and *B. cretica*). Our data showed that *B. cretica* made an extensive genetic contribution to all clades and groups in *B. oleracea*, ranging from 3.78% in LEE groups of cauliflower to 5.56% in cabbage, whereas *B. macrocarpa* contributed about 1.53% and other wild relatives contributed 0.94% on average (Fig. 2a). The distribution of these identical regions indicated that they are scattered across the entire genome, with short fragments of 5 kb in length occupying the majority of homologous sequences (Fig. 2b,c and Supplementary Figs. 6 and 7). Notably, genomic contributions varied among different wild accessions, ranging from 2.16% to 9.75% in *B. cretica* and from 0.92% to 2.47% in *B. macrocarpa*. The *B. cretica* accessions possessed the greatest number of identical regions with all *B. oleracea* subspecies (Supplementary Table 7). These results support *B. cretica* as the closest wild ancestor of cauliflower and suggest that it might be the origin of all *B. oleracea* subspecies[4]. To characterize the landscape of synteny within the cauliflower genome, we compared the genotype of the most similar *B. cretica* accession C0_0162 to the pseudo-ancestral genotype derived from a consensus of the LEE groups. We detected 4,996 candidate identical regions, ranging from 5 kb to 260 kb and harboring 5,980 genes. Gene ontology (GO) analysis revealed that these genes were overrepresented in lipid and fatty acid metabolism, including lipid transport (biological process (BP), GO:0006869), long-chain fatty acid metabolic process (BP, GO:0001676), long-chain fatty acid-CoA ligase activity (molecular function (MF), GO:0004467) and lipid transporter activity (MF, GO:0005319) (Fig. 2d).

## Genomic selection for curd formation in cauliflower

In *B. oleracea*, cauliflower has its own morphological and biological characteristics, including curd derived from specialized inflorescence meristems, plant height, biomass, and tolerance to biotic and abiotic stresses. Since cauliflower has been domesticated and cultivated worldwide, the genomic regions associated with its agronomic traits have changed substantially through continuous artificial selection, especially for the edible high-nutrient curd. To investigate the mechanism of curd biogenesis during cauliflower domestication, we merged clades 1 and 2 as an assumed 'Curdless' category, and clade 3 and the ROM group from clade 4 as a 'Green-curd' category, as well as the ELMC, LMC, EMC-1 and EMC-2 groups from clade 4 as a 'White-curd' category. In total, we identified 211 highly divergent genomic regions between the Curdless and Green-curd categories (defined as Curd-emergence) using the $F_{ST}$ method, and 185 between the Green-curd and White-curd categories (Curd-improvement). These divergent regions covered 50.7 Mb (8.92%; Curd-emergence) and 50.2 Mb (8.83%; Curd-improvement) of the C-8 (V2) genome, comprising 5,136 and 5,664 protein-coding genes, respectively (Supplementary Tables 8–11). GO analysis showed that the Curd-emergence genes were involved in maturation of 5.8S ribosomal RNA (BP, GO:0000460), cleavage involved in ribosomal RNA processing (BP, GO:0000469), ATP metabolic process (BP, GO:0046034) and preribosome (cellular component, GO:0030684). The Curd-improvement genes were enriched in protein maturation (BP, GO:0051604), plant epidermis development (BP, GO:0090558) and negative regulation of phosphorus metabolic process (BP, GO:0010563) (Supplementary Table 12).

Curd development occurs at the initial stage of flowering, during which the emerging primordia are transformed into curd-shaped inflorescences instead of floral organs[32]. To elucidate the underlying mechanisms of curd formation, we first collected all known flowering-related genes in *Arabidopsis* and then identified 519 homologs in the C-8 (V2) genome (Supplementary Table 13). Of these homologs, 55 and 61 flowering-related candidate genes resided in the significantly divergent genomic regions during the Curd-emergence and Curd-improvement processes, respectively (Fig. 3a,b and Supplementary Tables 14 and 15). The discrimination capacities of these genes showed successive declines in the above two processes, indicating that continuous artificial selection may have occurred throughout cauliflower domestication (Fig. 3c). Further investigation revealed that the upstream regulatory regions of three MADS-box genes, *CAL1*, *CAL2* (*AP1*) and *FUL2* (*AGL8.2*), varied between the Curdless and Green-curd categories (Fig. 3d), potentially affecting their function through transcriptional regulation. These findings are consistent with those of a previous study in *Arabidopsis* showing that *CAL* and *AP1* control the 'curd-like' phenotype, which arises from an abnormal inflorescence meristem[18]. More informatively, we found that the promoter region of *FUL2*, a gene controlling meristem arrest and lifespan in *Arabidopsis*[19], further differed between the Green-curd and White-curd categories. Tissue-specific transcriptome analysis showed that *CAL1*, *CAL2* and *FUL2* were indeed mainly expressed in cauliflower curd and floral organs (bud and flower) (Supplementary Fig. 8 and Table 16). To further verify divergent genomic regions related to curd formation, we performed bulked segregant analysis in two $F_2$ segregating populations derived from crossing of the Curdless and White-curd lines, each consisting of approximately 1,000 individuals. The differences (ΔSNP index) between the Curdless and White-curd bulks showed eight previously identified divergent genomic regions contributing to curd formation, containing *SEP3.3*, *CAL1*, *RPL18.2*, *TFL1.1* and *RPL3* on chromosome 3; *SEP3.1* on chromosome 5; *NAC071.1* and *FUL2* on chromosome 7; and *LIP1.2*, *FVE2*, *ABI2.1* and *WOX12.1* on chromosome 9 (Fig. 3e,f). To summarize, we propose a stepwise domestication of curd biogenesis containing two different sets of loci that may jointly give rise to cultivated White-curd.

We further analyzed the expression levels of these genes at the vegetative, curd initiation, curd expansion, curd premature and curd mature stages of curd furmation[10] and identified 21 potential curd-biogenesis-related genes that were differentially expressed during the Curd-emergence and Curd-improvement stages. In addition to the well-known genes *CAL1*, *CAL2*, *FUL2*, *TFL1.1*, *SEP3.1* and *SEP3.3*, whose homologs regulate floral organ development in *Arabidopsis*[13,19,33], we identified 15 genes comprising homologs of auxin-induced growth-related genes[34–36] (*WOX12.1*, *ARF9.1*, *HTA9.3* and *NAC071.1*), a circadian period-related gene[37] (*LIP1.2*), vernalization/autonomous-related genes[38–40] (*AGL19.2*, *FVE2* and *AGL6.1*), cytokinin- and abscisic acid-responsive genes[41] (*CYCD3;2.1*, *ABI1.2*, *ABI2.1* and *HB53.2*), housekeeping-related genes[42] (*RPL16B.4* and *RPL18.2*) and a regulatory-related gene (*RPL3*) (Fig. 3g and Supplementary Table 17). To understand the regulatory network responsible for curd formation, we constructed a panoramic view of regulatory events by integrating circadian clock, vernalization and autonomous pathways, as well as environmental signals, microRNAs and phytohormones including auxin, cytokinin, abscisic acid, brassinosteroids and gibberellic acid (Fig. 3h). In this analysis, multiple molecular interactions and environmental responses indicated that regulatory events during curd formation might be more complex than previously expected. However, the mechanisms and causal variations of these genes need to be further validated functionally.

## Genome-wide association studies of important agronomic traits

After continuous improvement, the cauliflower LEE groups have been bred into various edible varieties with diverse characteristic traits such as curd properties, resistance to pathogens, maturity and biomass. However, the genetic basis of most traits has not yet been elucidated in cauliflower. Therefore, to identify potential target genes or loci, we measured seven agronomic traits—stem height (SH), curd diameter (CD), curd height (CH), whole-plant mass (WPM), black rot resistance (BRR), color of curd branch (CCB) and insect resistance (IR)—and performed genome-wide association studies (GWAS) using 1.87 million SNPs from a panel composed of 691 cauliflower accessions. A total of nine dominant association signals were identified in the C-8 (V2) genome, and several candidate genes were speculated to be significantly associated with seven agronomic traits in cauliflower. These included *BOBO4G169050* (encoding an ENT domain-containing

**Fig. 3 | Genomic signatures and candidate genes involved in Curd-emergence and Curd-improvement. a,b,** Genome-wide screening of $F_{ST}$ differentiated signals for Curd-emergence (**a**) and Curd-improvement (**b**) during cauliflower domestication. Red dashed lines indicate the top 5% threshold set to select highly differentiated regions. Overall, 55 (Curd-emergence) and 61 (Curd-improvement) flowering-related genes located in the highly differentiated regions are indicated by gray arrows. The 21 genes showing altered expression profiles during curd development are labeled with their names. **c,** The discrimination capacity of differentiated flowering-related genes in the Curdless, Green-curd and White-curd categories (centerline, median; box limits, first and third quartiles; whiskers, 1.5× interquartile range). **d,** Nucleotide diversity (π) and read mapping diagrams for *CAL1*, *CAL2* and *FUL2* in each category. Red boxes indicate the promoter regions of target genes. **e,f,** Bulked segregant analysis (BSA) with $F_2$ populations developed by crossing the Curdless versus White-curd lines PQ409 (**e**) and PQ432 (**f**). Black dotted lines indicate the thresholds at 95% confidence level. Light purple blocks indicate BSA intervals overlapping with important $F_{ST}$ signals. **g,** Gene expression heatmap of differentially expressed genes at different stages of curd development. S0, vegetative; S1, curd initiation; S2, curd expansion; S3, curd premature; S4, curd mature. **h,** Integrated regulatory network showing potential mechanisms underlying curd biogenesis. This network covers the 21 candidate genes identified in this study as well as interacting genes, key flowering-related genes, phytohormones, microRNAs and environmental factors. White bars indicate 5 cm.

protein) and *BOB06G135460* (RING-type zinc-finger protein) for SH; *BOB03G039150* (elongation factor) and *BOB03G039160* (nonspecific serine/threonine protein kinase) for CD; *BOB04G016240* (unknown), *BOB04G016250* (ATP-dependent zinc metalloprotease FtsH) and *BOB08G004150* (TATA-box-binding protein) for CH, *BOB02G184480* (transcription repressor) for WPM; *BOB03G053850* (prokaryotic RING

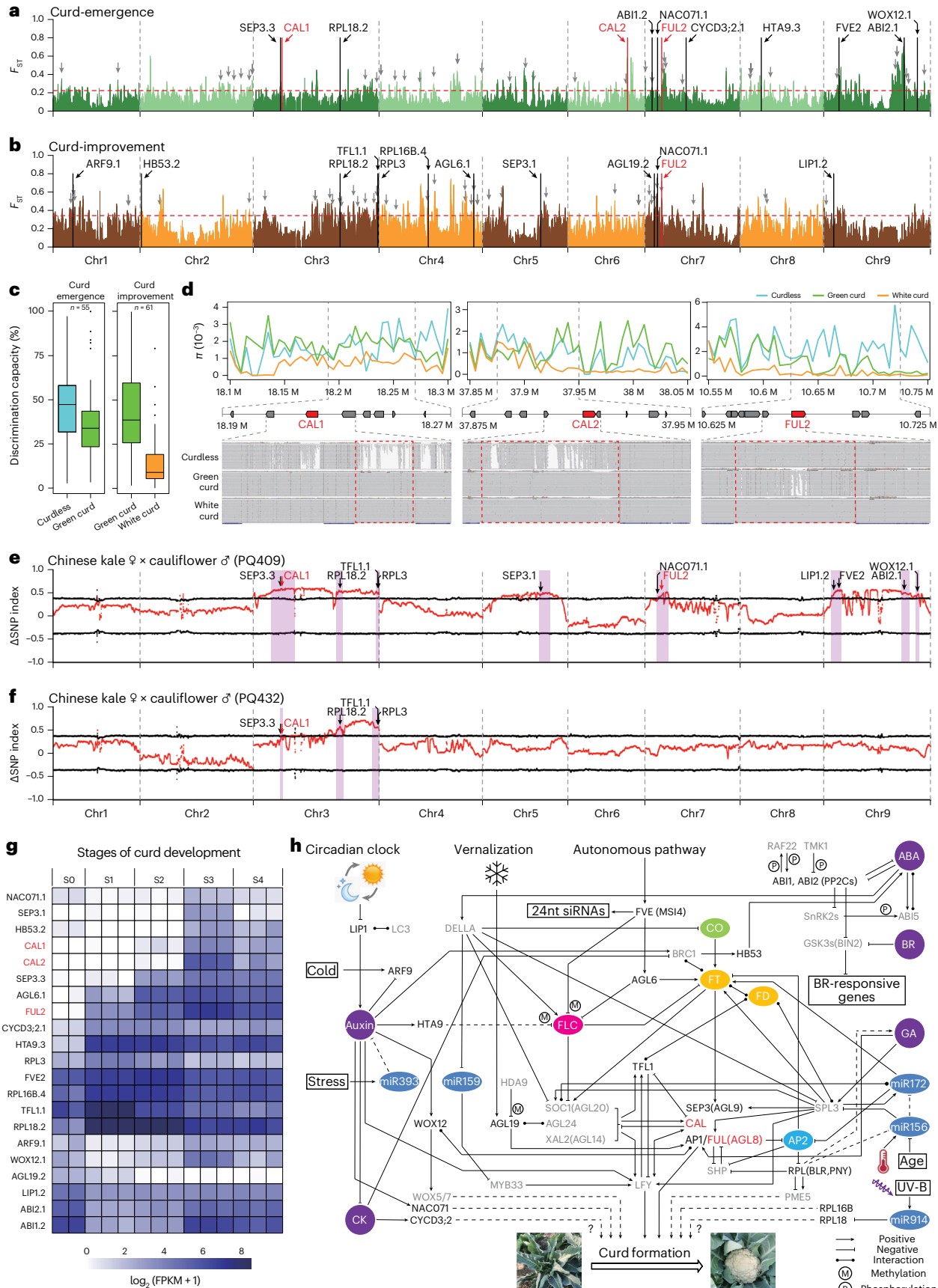

**Table 1 | GWAS-identified loci and candidate genes for important agronomic traits in cauliflower**

| Traits | Year | Peak position | −log$_{10}$(P) | Candidate genes |
|---|---|---|---|---|
| SH | 2019<br>2020 | Chr4:61010156<br>Chr6:47922380 | 6.63<br>6.95 | *BOB04G169050* (ENT domain-containing protein)<br>*BOB06G135460* (RING-type zinc-finger protein) |
| CD | 2020 | Chr3:14332232 | 8.28 | *BOB03G039150* (elongation factor), *BOB03G039160* (nonspecific serine/threonine protein kinase) |
| CH | 2019<br>2020 | Chr4:6787371<br>Chr8:1572454 | 6.34<br>6.83 | *BOB04G016240, BOB04G016250* (ATP-dependent zinc metalloprotease FtsH) *BOB08G004150* (TATA-box-binding protein) |
| WPM | 2019 | Chr2:71803792 | 7.20 | *BOB02G184480* (transcription repressor) |
| BRR | 2019 | Chr3:19497037 | 7.33 | *BOB03G053850* (prokaryotic RING finger family 4) |
| CCB | 2019 | Chr3:60340155 | 6.52 | *BOB03G161490* (DnaJ molecular chaperone) |
| IR | 2019 | Chr9:2162397 | 6.96 | *BOB09G004730* (protein kinase) |

finger family 4) for BRR; *BOB03G161490* (DnaJ molecular chaperone) for CCB; and *BOB09G004730* (protein kinase) for IR (Table 1 and Supplementary Fig. 9).

SH is an important agronomic trait that influences light capture, curd yield and the efficiency of mechanical picking of cauliflower (Fig. 4a). Phenotypic data of SH exhibited normal distributions in 2019 and 2020 (Fig. 4b,c). Correlation analysis indicated that SH has significant positive correlations with CD, CH and WPM traits (Supplementary Fig. 10). Our GWAS identified a strong association signal at the end of chromosome 6 for SH (2019, $P = 2.8 \times 10^{-7}$; 2020, $P = 1.1 \times 10^{-7}$) and CH (2020, $P = 2.5 \times 10^{-7}$) (Fig. 4d). Further analysis narrowed this interval to approximately 72 kb between 47.88 and 47.95 Mb; 12 protein-coding genes were located in this region based on the threshold value ($P = 1.0 \times 10^{-5}$) (Fig. 4e). Functional annotation and variant analysis revealed a RING-type zinc-finger gene (*BOB06G135460*) harboring one nonsynonymous SNP and a 3-bp deletion within its sixth exon that were present in most short-stem accessions (Fig. 4f). Haplotype analysis of this gene showed significant differences for SH, CD, CH and WPM traits in both 2019 and 2020 (Fig. 4g and Supplementary Fig. 11). The orthologs of this gene are widespread among monocots and dicots but exhibit divergent functions (Supplementary Fig. 12). For instance, a RING-type protein with E3 ubiquitin ligase activity (DA2) controls seed size by restricting cell proliferation in the maternal integuments in *Arabidopsis*[43], and another ortholog (GW2) regulates seed size[44] and leaf senescence[45] in rice. The expression of *BOB06G135460* dramatically increased at the vegetative and curd harvest stages in nine tall-stem accessions compared with nine short-stem accessions (Fig. 4h).

To further validate the function of *BOB06G135460*, we used a CRISPR–Cas9 editing strategy and generated three T$_0$ independently transformed lines (Fig. 4i). We found that the knockout lines (2.35–2.45 cm) had significantly shorter SH than the wild-type line (5.52 cm) (Fig. 4j). By contrast, the overexpression lines displayed obvious elongated SH (Supplementary Fig. 13). Microscopic observation showed that the thin-walled cortical cells in tall-stem accessions were significantly larger than those in short-stem accessions (Fig. 4k). Further analysis showed that the cell density (cell number per 500 μm$^2$) in tall-stem accessions was approximately three times lower than that in short-stem accessions (Fig. 4l,m), indicating that SH is mainly determined by the

size of cells in the cauliflower stem. Taken together, these results demonstrate that *BOB06G135460* positively regulates SH through affecting cell size during stem development, simultaneously affecting curd size and plant biomass in cauliflower.

## Discussion

*Brassica oleracea* has rich morphological diversity represented by highly specialized inflorescence, leaf, lateral bud and stem organs in its subspecies. Despite the attempts of numerous studies to untangle the origin and genetic relationships of *B. oleracea* populations, these details have remained unclear because of the frequent crossing and fully fertile progeny among wild and domesticated accessions. Taking advantage of large-scale sampling and high-density SNP markers, we carried out a comprehensive population genetic analysis and obtained a reasonable classification of *B. oleracea* composed of four major clades.

We confirmed that Chinese kale (clade 1) is the earliest divergent lineage, consistent with the fact that its cultivation history in China spans more than 1,000 years. Feral samples in clade 2 reflected the complicated nature of *B. oleracea* and made it difficult to clarify the evolutionary relationships. In addition, our results confirmed that the Aegean-endemic *B. cretica* is the closest wild ancestor of *B. oleracea*, although *B. cretica* individuals have unequal contributions owing to intraspecific diversity. Cauliflower (clade 4) is thought to possess a very narrow genetic background and have diverged less than 2,500 years ago[5]. However, its evolutionary history has never been thoroughly investigated because of the lack of wild germplasm resources and geographical origin information[5–8]. Herein, benefiting from large-scale sampling (726 accessions of cauliflower) and a whole-genome resequencing strategy, we divided the cauliflower population into five groups based on morphotypes and curd maturity levels. Among these, the ROM group has been traditionally classified as a type of cauliflower and displays a distinct curd morphotype, which is clearly different from broccoli and cauliflower curds in terms of color and shape. Our results also support the speculation that cauliflower directly evolved from broccoli. Importantly, we discovered the stepwise evolutionary route of cauliflower, from broccoli (clade 3) to the ROM group and next to the ELMC group, finally evolving into early-maturing cauliflower cultivars.

Curd development is a key concern in cauliflower breeding, as it affects yield and quality. Recent research in *Arabidopsis* has shown that Curd-emergence can be attributed to the combination of a few floral meristem determinants including *TFL1*, *LFY*, *CAL*, *AP1*, *SOC1* and *AGL24* (ref. 13). Cauliflower curd biogenesis was further illustrated using a batch of genes containing structural variants between the cauliflower and cabbage genomes[10]. In this study, we explored two steps of cauliflower domestication (Curd-emergence and Curd-improvement) and identified 21 candidate genes and their potential regulatory network based on expression profiles during curd development and found that *CAL1*, *CAL2/AP1* and *FUL2* might be key causal genes for curd formation. Our dataset will provide new routes for research into the genetic mechanisms of curd biogenesis and important resources for better understanding florescent development in crops.

In cauliflower, we found that SH was closely correlated with curd size and plant biomass. Previous studies had identified four quantitative trait loci (QTLs)[46] and multiple factors, including endogenous hormones[47] and gibberellin-related genes (*DELLA*[48] and *SOC1* (ref. 49)), that influenced stem elongation in *Brassica* plants. Nevertheless, no causal gene for stem elongation had been identified. Based on GWAS analysis, we discovered a RING-type zinc-finger protein-encoding gene, *BOB06G135460*, that controls stem elongation by influencing cell size. Its orthologous genes have been demonstrated to regulate seed size in *Arabidopsis*[43] and rice[44], suggesting that *BOB06G135460* could have versatile roles in regulating SH and associated traits in cauliflower. Notably, ~72–83% of haplotypes in short-stem cultivars could be explained, and

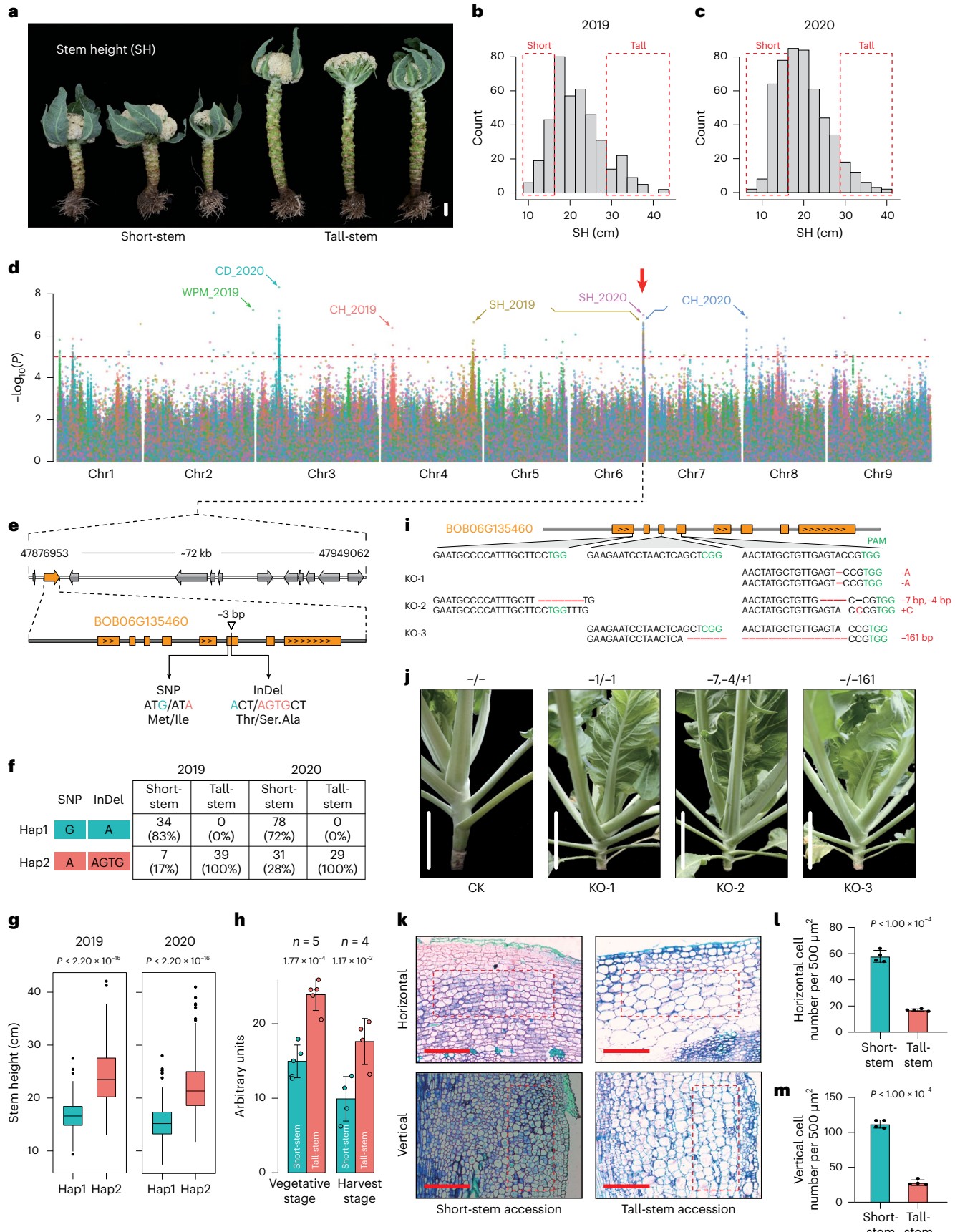

another GWAS peak on chromosome 4 (2019, $P = 2.4 \times 10^{-7}$) was detected, suggesting a minor quantitative trait locus (QTL) associated with SH (Fig. 4e). These results offer useful information for studying stem development and biomass regulation in *Brassica* plants. In addition, a few GWAS-identified loci and candidate genes responsible for important agronomic traits will facilitate cauliflower improvement in the future.

**Fig. 4 | GWAS-based dissection of SH trait and causative gene. a**, Morphology of short-stem and tall-stem cauliflower. Scale bar, 5 cm. **b**,**c**, Histograms of SH data in 2019 (**b**) and 2020 (**c**). Red boxes indicate the data of short-stem and tall-stem cauliflower, respectively, with thresholds set to ≤15 cm for short-stem and ≥30 cm for tall-stem. **d**, Overlapping Manhattan plots of SH, CD, CH and WPM traits. The red arrow indicates overlapping signals from SH and CH (Bonferroni correction). **e**, Schematic view of the target 72-kb region and candidate gene *BOB06G135460*. **f**, Statistics of haplotypes based on the nonsynonymous SNP and 3-bp deletion located in the sixth exon of *BOB06G135460*. **g**, Proportions of different haplotypes in SH in 2019 ($n_{Hap1} = 125$, $n_{Hap2} = 244$) and 2020 ($n_{Hap1} = 172$, $n_{Hap2} = 391$) (centerline, median; box limits, first and third quartiles; whiskers, 1.5×

interquartile range). **h**, Expression analysis of *BOB06G135460* for two haplotypes at the vegetative (84-day-old) and harvest (119-day-old) stages. Data are presented as means ± s.d. **i**, CRISPR–Cas9-induced mutations in *BOB06G135460* in $T_0$ plants. **j**, Plant morphologies of three knockout (KO) lines. White bars indicate 5 cm. **k**, Horizontal and vertical sections of stem tissues in short-stem and tall-stem accessions. Red scale bar, 500 μm. Red boxes indicate the regions of thin-walled cortical cells used for calculating cell numbers. **l**,**m**, Quantitative analysis of cell density for horizontal (**l**) and vertical (**m**) sections. Data are presented as mean ± s.e.m. ($n = 4$). In **g**, **h**, **l** and **m**, statistical significance was determined by two-sided Student's *t*-tests.

Collectively, we updated a high-quality and highly contiguous reference genome C-8 (V2) and implemented large-scale whole-genome resequencing in cauliflower, providing resources for gene mining and genome-guided breeding. Our findings shed light on the population structure and evolutionary history of *B. oleracea*. Moreover, candidate genes identified in this study related to curd formation and important agronomic traits will facilitate germplasm innovation in cauliflower.

## Online content

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

## Methods

### Genome assembly and annotation

In our previous study, a draft genome assembly and its annotation were reported using the elite cauliflower inbred line C-8 (ref. 9). Here, on the basis of existing raw data derived from PacBio (121×), Illumina (81×) and RNA-seq, we used complementary Hi-C library sequencing and Bionano optical mapping approaches to achieve a chromosomal-level genome assembly. Ten-day-old seedlings grown under greenhouse conditions were harvested and stored at −80 °C for subsequent experiments. The Hi-C library was constructed following the Proximo Hi-C plant protocol (Phase Genomics) with HindIII digestion, producing ~50.6 Gb raw reads (89×). To generate Bionano single-molecule maps, high-molecular-weight DNA with fragment distribution greater than 150 kb was isolated. Then, 300 ng of isolated DNA was incubated for 2 h at 37 °C with the Nt.BspQI enzyme for DNA nicking. After labeling of nicks using an IrysPrep Reagent Kit (Bionano Genomics) according to the manufacturer's instructions, the labeled DNA sample was loaded and imaged using the Bionano Saphyr platform (Bionano Genomics), producing ~98.4 Gb raw data with an average length of 270 kb.

Canu (v.2.2)[50] and the HERA pipeline (v.1.0)[51] were used for de novo genome assembly by producing contigs and merging repetitive regions with PacBio long reads and Illumina paired-end reads. During this process, Minimap2 (v.2.23)[52] and BWA (v.0.7.10-r789)[53] were employed for sequence alignment and overlap identification. Then, optical-map alignment and hybrid scaffolding were performed using the IrysView package (v.2.4.0.15879, Bionano Genomics) with a minimum length of 150 kb. Subsequently, scaffolds were further clustered by Hi-C data and 3D-DNA[54] with default parameters. After three rounds of base polishing with Pilon[55], the integrity of the final genome assembly (C-8, V2) was assessed with BUSCO (v.4.1.4)[56]. Analysis of genome-wide synteny was performed using SyRI (v.1.4)[57].

Before genome annotation, repeat analysis was accomplished by integrating de novo and homology-based methods and using Repeat-Modeler (v.2.0.1)[58], LTR_retriever (v.2.9.0)[59] and RepeatMasker (v.4.1.0)[60]; this included identification of interspersed transposable elements. A comprehensive pipeline for genome annotation was established by combining evidence from ab initio predictions, transcript mapping and cross-genome protein homology. In brief, tissue-specific RNA-seq data of cauliflower C-8 was cleaned and mapped onto the repeat-masked genome using HISAT2 (v.2.2.1)[61]. Coding sequences were assembled and recognized with TopHat2 (v.2.1.5)[62], Trinity (v.2.13.2)[63] and the PASA pipeline (v.2.4.1)[64]. Augustus (v.3.4.0)[65] and GeneMark-ES (v.3.67)[66] were used for ab initio gene predictions. Last, high-confidence gene models were integrated and summarized using the MAKER pipeline (v.2.31.11)[67].

### Plant materials and whole-genome resequencing

For whole-genome resequencing, 820 inbred lines of cauliflower and *B. oleracea* relatives were collected, and developed and stored in the Tianjin Kernel Vegetable Research Institute, Tianjin Academy of Agricultural Sciences. These lines included three wild accessions (one each of *B. macrocarpa*, *B. cretica* and *B. oleracea*), 53 Chinese kale accessions (var. *alboglabra*), nine kohlrabi accessions (var. *gongylodes*), two lacinato kale accessions (var. *palmifolia*), three curly kale accessions (var. *sabellica*), 11 Brussels sprouts accessions (var. *gemmifera*), two savoy cabbage accessions (var. *capitata*), three kale accessions (var. *acephala*), five cabbage accessions (var. *capitata*), 20 broccoli accessions (var. *italica*), 16 Romanesco cauliflower accessions (var. *botrytis*) and 693 cauliflower accessions (var. *botrytis*) (Supplementary Table 3). This collection had widespread origins and diverse genetic backgrounds, exhibiting abundant biological and morphological variations in traits such as maturity, biomass, disease resistance and curd characteristics.

Young leaves from 25-day-old seedlings of these accessions were subjected to genomic DNA isolation using a modified cetyl-trimethylammonium bromide method[68]. The PE150 strategy was used for library construction and deep sequencing on an Illumina NovaSeq 6000 platform at Novogene (Beijing, China), producing ~6.5 Tb raw reads corresponding to approximately 14× genomic depth for each sample. In addition, 151 sets of genome resequencing raw data were downloaded from the NCBI Sequence Read Archive (SRA) database (PRJNA217459, PRJNA301390, PRJNA312457, PRJNA320480, PRJNA428769, PRJNA470925 and PRJNA516907) (Supplementary Table 3).

### Sequence alignment and variant calling

First, raw reads were filtered using the Fastp program (v.0.12.4)[69] with default parameters. Clean reads for each sample were aligned onto the C-8 (V2) genome with the 'mem' algorithm in BWA (v.0.7.10-r789)[53]. SAMtools (v.1.14)[70] was then used to convert the format of SAM files, sort BAM files and filter mapping quality with the '-q 30' parameter. The Genome Analysis Toolkit (GATK, v.4.1.4.1)[71] modules MarkDuplicates and ValidateSamFile were used to remove duplicates and validate the file integrity, respectively. To improve variant calling efficiency, the genome was split into individual chromosomes for parallel calculation. For each chromosome of each sample, GATK HaplotypeCaller was used in -ERC GVCF mode to generate original GVCF files. Subsequently, the CombineGVCFs, GenotypeGVCFs, SelectVariants and VariantFiltration modules were applied in turn for SNP and InDel calling (SNPs: −filter-expression 'QD < 2.0 || MQ < 40.0 || FS > 60.0 || MQRankSum < −12.5 || ReadPosRankSum < −8.0' −cluster-size 3 −cluster-window-size 10; InDels: −filter-expression 'QD < 2.0 || FS > 200.0 || ReadPosRank-Sum < −20.0'). The SNPs or InDels that passed the screening criteria were extracted and gathered as high-confidence variants. Finally, the whole set of variants was annotated using SnpEff (v.4.3t)[72] with default parameters.

### Phylogenetic and population structure analyses

Considering that 4d-SNPs are under less selective pressure and can reliably reflect population structure and demography, we selected 4d-SNPs with a minor allele frequency greater than 0.05 and missing rate less than 20% as neutral or near-neutral SNPs. As a result, 69,275 4d-SNPs were obtained and subjected to construction of an ML phylogenetic tree using FastTree (v.2.1.11)[73]. Population structure was analyzed using the ADMIXTURE (v.1.3.0)[74] program with the same set of SNPs. For PCA, PLINK (v.1.90b5.3)[75] was utilized with parameters −geno 0.05, −hwe 0.0001 and −maf 0.05 for SNP filtration. PCA was performed on a subset of 1,564 SNPs using genome-wide complex trait analysis (GCTA, v.1.26.0)[76]. Population fixation statistics ($F_{ST}$) and genetic diversity ($\pi$) were calculated using VCFtools (v.0.1.16)[77] based on the whole set of SNPs. The $\pi$ levels were measured for each 100-kb window, and $F_{ST}$ values were estimated for 50-kb sliding windows with a step size of 5 kb. The average values of $\pi$ and $F_{ST}$ across the whole genome were designated as the final values for each clade or group. LD decay was calculated for all pairs of SNPs within 1000 kb using PopLDdecay (v.3.41)[78] with parameters -MaxDist 1000, -Het 0.1 and -Miss 0.1. Average $r^2$ values in a bin of 100 bp against the physical distance of pairwise bins were illustrated. Inbreeding coefficients were computed using PLINK[75] and GCTA software[76] with the command '-ibc'.

### Ancestral inference

Synteny analysis was carried out between the 22 wild accessions at the root of the phylogenetic tree and each *B. oleracea* subspecies. Briefly, the consensus genotype of a specified clade or group was reconstructed by selecting the most common allele composition across each individual. Consecutive 5-kb sliding windows were set to compare the identity between a wild accession and the inferred ancestral genotype at each SNP site. Only the windows with at least five shared SNPs and similarity greater than 96% were defined as syntenic regions and reserved for visualization with the RIdeogram package (v.0.2.2)[79].

## Identification of differentiated regions

With the aid of a variance component approach using the Hierfstat R package (v.0.5.10)[80], $F_{ST}$ values were estimated for 100-kb sliding windows with a step size of 10 kb. Sliding windows with the top 5% $F_{ST}$ values were initially selected. After merging neighboring windows, fragments were further merged into one region if the distance between two fragments was less than 100 kb. The final merged regions were considered to be highly diverged regions between different groups.

## Identification of flowering-related genes and genotype analysis

A comprehensive literature review was carried out to identify flowering-related genes in plants[10,13,81]. Following specific BLASTP thresholds (mutual coverage >70%, sequence identity >75% and mismatches/coverage <25%), 519 homologous genes with potential roles in curd development were identified in the cauliflower genome C-8 (V2). For each target gene, SNPs located in the gene body were connected into an assumed sequence, which was deemed to be its own genotype. For each group, the most abundant genotype was regarded as the representative genotype. Discrimination capacity was calculated by dividing the number of different genotypes by the total number of individuals in a certain group. GO enrichment analysis of the sweep genes was carried out using R package topGO (v.2.36.0)[82].

## Transcriptomic analysis

A total of 132 sets of *B. oleracea* tissue-specific RNA-seq data were downloaded from the SRA database (PRJNA183713, PRJNA227258, PRJNA231628, PRJNA289196, PRJNA292848, PRJNA297049, PRJNA428769, PRJNA489323, PRJNA516113, PRJNA525713, PRJNA546441, PRJNA548819, PRJNA633027, PRJNA683970). These datasets included RNA-seq data from root, stem, leaf, bud, flower and silique, as well as from curd organs (Supplementary Table 16). Transcriptome data of different curd developmental stages were also downloaded from the SRA database (PRJNA546441) (Supplementary Table 17). Fastq-dump (v.2.11.2) in the SRA Toolkit[83] and the Fastp program (v.0.12.4)[69] were used for format conversion and read cleaning. HISAT2 (v.2.2.1)[61] and the Cufflinks suite (v.2.2.1)[84] were used to estimate fragments per kilobase of transcript per million mapped reads (FPKM) values for each gene. Heatmaps were constructed using the R package pheatmap (v.1.0.12)[85] with $\log_2$ (FPKM + 1) values of selected genes.

## Planting and phenotyping

Curd is the specialized organ composed of enlarged and developmentally arrested inflorescence or floral meristems in cauliflower and broccoli. In this study, seven important agronomic traits were analyzed, comprising SH, CD, CH, WPM, BRR, CCB and IR. These traits were measured in two successive years from plants grown in two separate geographic locations: from the Baodi district of Tianjin municipality, China, in 2019, and from Hebei province, China, in 2020. The recording standards for the phenotype data refer to the description guidelines for germplasm resources of cauliflower and broccoli in China[86]. A scatterplot matrix with correlation values was produced using the ggpairs function in the GGally R package.

## Genome-wide association studies

SNP filtration was set as major allele frequency >0.05 and missing rate <0.2. As a result, 1,873,097 SNPs were qualified across cauliflower populations and used for GWAS with GAPIT3 (v.3.1.0)[87] with a mixed linear model. The significance threshold was set as $P = 1 \times 10^{-5}$. For phenotypic data, 450 accessions in 2019 and 607 accessions in 2020 were successfully collected. Downy mildew resistance was assessed using at least 20 individuals per accession. Other traits were measured in duplicate on five individuals per accession.

## Quantitative real-time PCR

To verify the expression of the target gene *BOB06G135460*, main stem tissues of 18 accessions and corresponding genotypes were sampled at the vegetative (84-day-old) and harvest (119-day-old) stages. Total RNA was isolated with an Eastep Super Total RNA Extraction Kit (Promega, LS1040) and used to synthesize first-strand cDNA with a PrimeScript RT Reagent Kit (TaKaRa, RR037A) according to the manufacturer's protocols (Supplementary Table 18). All quantitative real-time PCR reactions were performed using a TB Green Premix Ex Taq II kit (Takara, RR820A) in a LightCycler 480 II system (Roche Diagnostics) with reference gene *Actin* (*BOB02G179850*) as an internal control. The relative expression levels were calculated as $2^{-(CT\,target-CT\,control)} \times 1,000$ in arbitrary units.

## Cytological analysis

Phloem tissues were fixed in FAA (50% ethanol/formaldehyde/glacial acetic acid, 90:5:5) for 24 h then subjected to paraffin embedding and slicing as previously described[88]. Slices were stained with 0.5% tolonium chloride and photographed using a fluorescence microscope (VIYEE V5800, China). Three biological replicates derived from different cultivars (62 days old) were used for both short-stem and tall-stem samples. For each microscopic picture, four $500 \times 500\,\mu m^2$ squares were randomly selected to calculate the number of cortical thin-walled cells.

## Bulked segregant analysis

We created two $F_2$ populations, each consisting of about 1,000 individuals, using Chinese kale (PQ435) × cauliflower (PQ409) and Chinese kale (PQ435) × cauliflower (PQ432), planted in the spring of 2023 in the experimental bases of Zhangjiakou Academy of Agricultural Sciences (Zhangbei, Hebei province, China) and Tianjin Academy of Agricultural Sciences (Wuqing district, Tianjin municipality, China). Bulk DNA samples were collected by mixing equal amounts of DNA from 20 individuals with cauliflower-like phenotypes and 20 individuals with Chinese kale-like phenotypes, respectively. Roughly 20× raw data for each parent and 50× for each bulk sample were generated on the Illumina NovaSeq 6000 platform. BWA[53], SAMtools[70] and BCFtools were used for genome mapping and SNP calling. Only high-quality SNPs with base quality value >30 and mapping quality value >30 were retained for further analysis. SNP index and Δ(SNP index) parameters were calculated to identify candidate regions using a 1,000 kb sliding window with a step size of 10 kb. Combined with the statistical confidence intervals of the ΔSNP index under the null hypothesis of no QTLs, 95% confidence intervals of the ΔSNP index were finally extracted for each position[89].

## Vector construction and plant transformation

Overexpression and CRISPR/Cas9-mediated knockout were performed for functional validation of *BOB06G135460*. The full-length cDNA clone was integrated into the pCAMBIA3301 vector through RNA isolation and reverse transcription using the stem tissue of a high-stem genotype accession. The highly specific guide RNAs located in the exonic regions of *BOB06G135460* were integrated into the pCBC-DT1T2 and pKSE401 vectors for gene editing. *Agrobacterium tumefaciens*-mediated hypocotyl transformation was conducted as previously described[90]. The FQ-38 inbred line was used as the transformation receptor.

## Statistical analysis

Statistical significance was determined by two-sided Student's *t*-tests.

## Reporting summary

Further information on research design is available in the Nature Portfolio Reporting Summary linked to this article.

## Data availability

Hi-C raw reads (SRR18307894 and SRR18307895) and Bionano CMAP file PRJNA516113 have been deposited in the NCBI SRA database. The genome assembly of cauliflower C-8 (V2) has been deposited at DDBJ/

ENA/GenBank (https://www.ncbi.nlm.nih.gov/nucleotide) under accession JAMKOK000000000 and at the Genome Warehouse of the National Genomics Data Center (NGDC, https://ngdc.cncb.ac.cn/gwh) under accession GWHBJSH00000000. Resequencing raw reads derived from 820 accessions of cauliflower and *B. oleracea* relatives have been deposited in the SRA database under BioProject accession PRJNA794342. The raw reads of bulked segregant sequencing have been deposited in the SRA database under BioProject accession PRJNA1082923 and at the Genome Sequence Archive NGDC database (CRA012694).

## Code availability
Custom scripts and codes used in this study are provided at Zenodo (https://doi.org/10.5281/zenodo.10824481)[91] and GitHub (https://github.com/ChenRui-TAAS/Cauliflower_Resequencing). Software and tools used are described in the Methods and Reporting Summary.

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

## Acknowledgements

We thank X. Wang (Institute of Vegetables and Flowers, Chinese Academy of Agricultural Sciences) for critical comments, and M. Avramakis (University of Crete, Greece) for the pictures of *B. cretica*. This work was supported by funding from the Modern Agro-Industry Technology Research System of China (CARS-23-A-04 to D.S.), the National Key Research and Development Program of China (2022YFF1003000 to X.Z. and 2023YFF1000100 to T.L.), the 111 Project (B17043 to T.L.), the Construction of Beijing Science and Technology Innovation and Service Capacity in Top Subjects (CEFF-PXM2019_014207_000032 to T.L.), Hunan Youth Science and Technology Talent Project (2022RC1017 to K.C.), '131' innovative team construction project of Tianjin (201923 to X.Y.), the Vegetable Modern Agro-Industry Technology Research System of Tianjin (ITTVRS2017004 to X.Y.), the National Natural Science Foundation of China (31671964 to R.C., 32002042 to X.Z. and 32302579 to Yingxia Yang), the Natural Science Foundation of Tianjin (22JCYBJC00190 to Yingxia Yang, 23JCYBJC00770 to M.L. and 23JCQNJC01040 to Q.W.) and the Innovation Research and Experiment Program for Youth Scholar of Tianjin Academy of Agricultural Sciences (2021023 to Q.W. and 2022017 to Yingxia Yang).

## Author contributions

T.L., R.C. and D.S. designed studies and initiated this project. D.S., X.Y., X.Z., X.G. and P.F.S. contributed to the collection of *B. oleracea* accessions. X.Y., X.Z., G.Z., M.W., L.D. and T.X. planted accessions, prepared the samples and performed phenotyping. T.L., K.C. and X.S. assembled and annotated the genome. T.L., R.C., K.C., Yingxia Yang, X.S., M.L., Q.W., G.Z., Yuyao Yang and G.J. performed the bioinformatics analyses. R.C., Yingxia Yang, M.W., H.L., H.Z. and Y.G. designed and performed the molecular experiments. Y.L. and T.X. performed the cytological experiments. T.L., R.C., D.S., K.C., X.Y. and X.Z. wrote and/or revised the manuscript. All authors read and approved the manuscript.

## Competing interests

The authors declare no competing interests.

## Additional information

**Correspondence and requests for materials** should be addressed to Rui Chen, Tao Lin or Deling Sun.

# Reporting Summary

## Statistics

For all statistical analyses, confirm that the following items are present in the figure legend, table legend, main text, or Methods section.

| n/a | Confirmed | |
|---|---|---|
| ☐ | ☒ | The exact sample size (*n*) for each experimental group/condition, given as a discrete number and unit of measurement |
| ☐ | ☒ | A statement on whether measurements were taken from distinct samples or whether the same sample was measured repeatedly |
| ☐ | ☒ | The statistical test(s) used AND whether they are one- or two-sided<br>*Only common tests should be described solely by name; describe more complex techniques in the Methods section.* |
| ☒ | ☐ | A description of all covariates tested |
| ☒ | ☐ | A description of any assumptions or corrections, such as tests of normality and adjustment for multiple comparisons |
| ☐ | ☒ | A full description of the statistical parameters including central tendency (e.g. means) or other basic estimates (e.g. regression coefficient) AND variation (e.g. standard deviation) or associated estimates of uncertainty (e.g. confidence intervals) |
| ☐ | ☒ | For null hypothesis testing, the test statistic (e.g. *F*, *t*, *r*) with confidence intervals, effect sizes, degrees of freedom and *P* value noted<br>*Give P values as exact values whenever suitable.* |
| ☒ | ☐ | For Bayesian analysis, information on the choice of priors and Markov chain Monte Carlo settings |
| ☒ | ☐ | For hierarchical and complex designs, identification of the appropriate level for tests and full reporting of outcomes |
| ☒ | ☐ | Estimates of effect sizes (e.g. Cohen's *d*, Pearson's *r*), indicating how they were calculated |

*Our web collection on statistics for biologists contains articles on many of the points above.*

## Software and code

Policy information about availability of computer code

| Data collection | Sequencing platforms used to generate the raw data are listed as followed: Illumina NovaSeq and Bionano Saphyr. |
|---|---|
| Data analysis | Detailed description for all the softwares used for analysis have been provided in the Methods section. The tools and software used in this study: Canu (v2.2), HERA pipeline (v1.0), Minimap2 (v2.23), BWA (v0.7.10-r789), IrysView package (v2.4.0.15879), BUSCO (v4.1.4), SyRI (v1.4), RepeatModeler (v2.0.1), LTR_retriever (v2.9.0), RepeatMasker (v4.1.0), HISAT2 (v2.2.1), Tophat2 (v2.1.5), Trinity (v2.13.2), PASA pipeline (v2.4.1), Augustus (v3.4.0), GeneMark-ES (v3.67), MAKER pipeline (v2.31.11), Fastp (v0.12.4), Samtools (v1.14), GATK (v4.1.4.1), SnpEff (v4.3t), FastTree (v2.1.11), ADMIXTURE (v1.3.0), PLINK (v1.90b5.3), GCTA (v1.26.0), VCFtools (v0.1.16), PopLDdecay (v3.41), BLASTP (v2.7.1+), R package GGally (v2.1.1), RIdeogram package (v0.2.2), R package Hierfstat (v0.5.10), R package topGO (v2.36.0), Fastq-dump (v2.11.2), Cufflinks (v2.2.1), R package pheatmap (v1.0.12), and GAPIT3 (v3.1.0). Custom scripts and codes used in this study are provided at GitHub (https://github.com/ChenRui-TAAS/Cauliflower_Resequencing). |

For manuscripts utilizing custom algorithms or software that are central to the research but not yet described in published literature, software must be made available to editors and reviewers. We strongly encourage code deposition in a community repository (e.g. GitHub). See the Nature Portfolio guidelines for submitting code & software for further information.

# Data

Policy information about <u>availability of data</u>

All manuscripts must include a <u>data availability statement</u>. This statement should provide the following information, where applicable:
- Accession codes, unique identifiers, or web links for publicly available datasets
- A description of any restrictions on data availability
- For clinical datasets or third party data, please ensure that the statement adheres to our <u>policy</u>

Hi-C raw reads (SRR18307894 and SRR18307895), Bionano cmap file (SUPPF_0000004264) have been deposited in the NCBI SRA database. The whole genome sequence and annotation data have been deposited at DDBJ/ENA/GenBank under the accession JAMKOK000000000 and Genome Warehouse in National Genomics Data Center (NGDC) under the accession GWHBJSH00000000. Resequencing raw reads derived from 820 accessions of cauliflower and B. oleracea relatives have been deposited in the SRA database under a BioProject accession (PRJNA794342). The raw reads of bulked segregant sequencing have been deposited in the SRA database under a BioProject accession (PRJNA1082923) and the Genome Sequence Archive (GSA) database, NGDC (CRA012694).

# Research involving human participants, their data, or biological material

Policy information about studies with <u>human participants or human data</u>. See also policy information about <u>sex, gender (identity/presentation), and sexual orientation</u> and <u>race, ethnicity and racism</u>.

| | |
|---|---|
| Reporting on sex and gender | N/A |
| Reporting on race, ethnicity, or other socially relevant groupings | N/A |
| Population characteristics | N/A |
| Recruitment | N/A |
| Ethics oversight | N/A |

Note that full information on the approval of the study protocol must also be provided in the manuscript.

# Field-specific reporting

Please select the one below that is the best fit for your research. If you are not sure, read the appropriate sections before making your selection.

☒ Life sciences        ☐ Behavioural & social sciences        ☐ Ecological, evolutionary & environmental sciences

For a reference copy of the document with all sections, see <u>nature.com/documents/nr-reporting-summary-flat.pdf</u>

# Life sciences study design

All studies must disclose on these points even when the disclosure is negative.

| | |
|---|---|
| Sample size | 820 samples of cauliflower and B. oleracea relatives were selected for whole genome resequencing, including three wild accessions (each one of B. macrocarpa, B. cretica and B. oleracea), 53 Chinese kale accessions (var. alboglabra), nine kohlrabi accessions (var. gongylodes), two lacinato kale accessions (var. palmifolia), three curly kale accessions (var. sabellica), 11 Brussels sprouts accessions (var. gemmifera), two savoy cabbage accessions (var. capitata), three kale accessions (var. acephala), five cabbage accessions (var. capitata), 20 broccoli accessions (var. italica), 16 Romanesco cauliflower accessions (var. botrytis), and 693 cauliflower accessions (var. botrytis). This collection has widespread origins and diverse genetic backgrounds, exhibiting abundant biological and morphological variations in traits such as maturity, biomass, disease resistance, and curd characteristics, which is sufficient to represent the B. oleracea populations. |
| Data exclusions | No data was excluded from the analyses. |
| Replication | For qRT-PCR analysis, three independent experiments were performed. Three to five sampling replicates were used for measuring the phenotypes of agronomic traits in cauliflower. All replications were successful and were used. |
| Randomization | For each accession, the sampling process for genome DNA sequencing was randomly conducted. |
| Blinding | Blinding is not necessary for genome resequencing, since the investigators know which accessions they were dealing with. |

# Behavioural & social sciences study design

All studies must disclose on these points even when the disclosure is negative.

| | |
|---|---|
| Study description | *Briefly describe the study type including whether data are quantitative, qualitative, or mixed-methods (e.g. qualitative cross-sectional, quantitative experimental, mixed-methods case study).* |
| Research sample | *State the research sample (e.g. Harvard university undergraduates, villagers in rural India) and provide relevant demographic information (e.g. age, sex) and indicate whether the sample is representative. Provide a rationale for the study sample chosen. For studies involving existing datasets, please describe the dataset and source.* |
| Sampling strategy | *Describe the sampling procedure (e.g. random, snowball, stratified, convenience). Describe the statistical methods that were used to predetermine sample size OR if no sample-size calculation was performed, describe how sample sizes were chosen and provide a rationale for why these sample sizes are sufficient. For qualitative data, please indicate whether data saturation was considered, and what criteria were used to decide that no further sampling was needed.* |
| Data collection | *Provide details about the data collection procedure, including the instruments or devices used to record the data (e.g. pen and paper, computer, eye tracker, video or audio equipment) whether anyone was present besides the participant(s) and the researcher, and whether the researcher was blind to experimental condition and/or the study hypothesis during data collection.* |
| Timing | *Indicate the start and stop dates of data collection. If there is a gap between collection periods, state the dates for each sample cohort.* |
| Data exclusions | *If no data were excluded from the analyses, state so OR if data were excluded, provide the exact number of exclusions and the rationale behind them, indicating whether exclusion criteria were pre-established.* |
| Non-participation | *State how many participants dropped out/declined participation and the reason(s) given OR provide response rate OR state that no participants dropped out/declined participation.* |
| Randomization | *If participants were not allocated into experimental groups, state so OR describe how participants were allocated to groups, and if allocation was not random, describe how covariates were controlled.* |

# Ecological, evolutionary & environmental sciences study design

All studies must disclose on these points even when the disclosure is negative.

| | |
|---|---|
| Study description | *Briefly describe the study. For quantitative data include treatment factors and interactions, design structure (e.g. factorial, nested, hierarchical), nature and number of experimental units and replicates.* |
| Research sample | *Describe the research sample (e.g. a group of tagged Passer domesticus, all Stenocereus thurberi within Organ Pipe Cactus National Monument), and provide a rationale for the sample choice. When relevant, describe the organism taxa, source, sex, age range and any manipulations. State what population the sample is meant to represent when applicable. For studies involving existing datasets, describe the data and its source.* |
| Sampling strategy | *Note the sampling procedure. Describe the statistical methods that were used to predetermine sample size OR if no sample-size calculation was performed, describe how sample sizes were chosen and provide a rationale for why these sample sizes are sufficient.* |
| Data collection | *Describe the data collection procedure, including who recorded the data and how.* |
| Timing and spatial scale | *Indicate the start and stop dates of data collection, noting the frequency and periodicity of sampling and providing a rationale for these choices. If there is a gap between collection periods, state the dates for each sample cohort. Specify the spatial scale from which the data are taken* |
| Data exclusions | *If no data were excluded from the analyses, state so OR if data were excluded, describe the exclusions and the rationale behind them, indicating whether exclusion criteria were pre-established.* |
| Reproducibility | *Describe the measures taken to verify the reproducibility of experimental findings. For each experiment, note whether any attempts to repeat the experiment failed OR state that all attempts to repeat the experiment were successful.* |
| Randomization | *Describe how samples/organisms/participants were allocated into groups. If allocation was not random, describe how covariates were controlled. If this is not relevant to your study, explain why.* |
| Blinding | *Describe the extent of blinding used during data acquisition and analysis. If blinding was not possible, describe why OR explain why blinding was not relevant to your study.* |

Did the study involve field work? 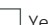 Yes 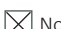 No

# Field work, collection and transport

| | |
|---|---|
| Field conditions | *Describe the study conditions for field work, providing relevant parameters (e.g. temperature, rainfall).* |
| Location | *State the location of the sampling or experiment, providing relevant parameters (e.g. latitude and longitude, elevation, water depth).* |
| Access & import/export | *Describe the efforts you have made to access habitats and to collect and import/export your samples in a responsible manner and in compliance with local, national and international laws, noting any permits that were obtained (give the name of the issuing authority, the date of issue, and any identifying information).* |
| Disturbance | *Describe any disturbance caused by the study and how it was minimized.* |

# Reporting for specific materials, systems and methods

We require information from authors about some types of materials, experimental systems and methods used in many studies. Here, indicate whether each material, system or method listed is relevant to your study. If you are not sure if a list item applies to your research, read the appropriate section before selecting a response.

### Materials & experimental systems

| n/a | Involved in the study |
|---|---|
| ☒ | Antibodies |
| ☒ | Eukaryotic cell lines |
| ☒ | Palaeontology and archaeology |
| ☒ | Animals and other organisms |
| ☒ | Clinical data |
| ☒ | Dual use research of concern |
| ☐ | ☒ Plants |

### Methods

| n/a | Involved in the study |
|---|---|
| ☒ | ChIP-seq |
| ☒ | Flow cytometry |
| ☒ | MRI-based neuroimaging |

## Antibodies

| | |
|---|---|
| Antibodies used | *Describe all antibodies used in the study; as applicable, provide supplier name, catalog number, clone name, and lot number.* |
| Validation | *Describe the validation of each primary antibody for the species and application, noting any validation statements on the manufacturer's website, relevant citations, antibody profiles in online databases, or data provided in the manuscript.* |

## Eukaryotic cell lines

Policy information about cell lines and Sex and Gender in Research

| | |
|---|---|
| Cell line source(s) | *State the source of each cell line used and the sex of all primary cell lines and cells derived from human participants or vertebrate models.* |
| Authentication | *Describe the authentication procedures for each cell line used OR declare that none of the cell lines used were authenticated.* |
| Mycoplasma contamination | *Confirm that all cell lines tested negative for mycoplasma contamination OR describe the results of the testing for mycoplasma contamination OR declare that the cell lines were not tested for mycoplasma contamination.* |
| Commonly misidentified lines (See ICLAC register) | *Name any commonly misidentified cell lines used in the study and provide a rationale for their use.* |

## Palaeontology and Archaeology

| | |
|---|---|
| Specimen provenance | *Provide provenance information for specimens and describe permits that were obtained for the work (including the name of the issuing authority, the date of issue, and any identifying information). Permits should encompass collection and, where applicable, export.* |
| Specimen deposition | *Indicate where the specimens have been deposited to permit free access by other researchers.* |

| Dating methods | *If new dates are provided, describe how they were obtained (e.g. collection, storage, sample pretreatment and measurement), where they were obtained (i.e. lab name), the calibration program and the protocol for quality assurance OR state that no new dates are provided.* |
|---|---|

☐ Tick this box to confirm that the raw and calibrated dates are available in the paper or in Supplementary Information.

| Ethics oversight | *Identify the organization(s) that approved or provided guidance on the study protocol, OR state that no ethical approval or guidance was required and explain why not.* |
|---|---|

Note that full information on the approval of the study protocol must also be provided in the manuscript.

# Animals and other research organisms

Policy information about studies involving animals; ARRIVE guidelines recommended for reporting animal research, and Sex and Gender in Research

| Laboratory animals | *For laboratory animals, report species, strain and age OR state that the study did not involve laboratory animals.* |
|---|---|
| Wild animals | *Provide details on animals observed in or captured in the field; report species and age where possible. Describe how animals were caught and transported and what happened to captive animals after the study (if killed, explain why and describe method; if released, say where and when) OR state that the study did not involve wild animals.* |
| Reporting on sex | *Indicate if findings apply to only one sex; describe whether sex was considered in study design, methods used for assigning sex. Provide data disaggregated for sex where this information has been collected in the source data as appropriate; provide overall numbers in this Reporting Summary. Please state if this information has not been collected. Report sex-based analyses where performed, justify reasons for lack of sex-based analysis.* |
| Field-collected samples | *For laboratory work with field-collected samples, describe all relevant parameters such as housing, maintenance, temperature, photoperiod and end-of-experiment protocol OR state that the study did not involve samples collected from the field.* |
| Ethics oversight | *Identify the organization(s) that approved or provided guidance on the study protocol, OR state that no ethical approval or guidance was required and explain why not.* |

Note that full information on the approval of the study protocol must also be provided in the manuscript.

# Clinical data

Policy information about clinical studies
All manuscripts should comply with the ICMJE guidelines for publication of clinical research and a completed CONSORT checklist must be included with all submissions.

| Clinical trial registration | *Provide the trial registration number from ClinicalTrials.gov or an equivalent agency.* |
|---|---|
| Study protocol | *Note where the full trial protocol can be accessed OR if not available, explain why.* |
| Data collection | *Describe the settings and locales of data collection, noting the time periods of recruitment and data collection.* |
| Outcomes | *Describe how you pre-defined primary and secondary outcome measures and how you assessed these measures.* |

# Dual use research of concern

Policy information about dual use research of concern

## Hazards

Could the accidental, deliberate or reckless misuse of agents or technologies generated in the work, or the application of information presented in the manuscript, pose a threat to:

| No | Yes | |
|---|---|---|
| ☐ | ☐ | Public health |
| ☐ | ☐ | National security |
| ☐ | ☐ | Crops and/or livestock |
| ☐ | ☐ | Ecosystems |
| ☐ | ☐ | Any other significant area |

## Experiments of concern

Does the work involve any of these experiments of concern:

| No | Yes | |
|----|-----|---|
| ☐ | ☐ | Demonstrate how to render a vaccine ineffective |
| ☐ | ☐ | Confer resistance to therapeutically useful antibiotics or antiviral agents |
| ☐ | ☐ | Enhance the virulence of a pathogen or render a nonpathogen virulent |
| ☐ | ☐ | Increase transmissibility of a pathogen |
| ☐ | ☐ | Alter the host range of a pathogen |
| ☐ | ☐ | Enable evasion of diagnostic/detection modalities |
| ☐ | ☐ | Enable the weaponization of a biological agent or toxin |
| ☐ | ☐ | Any other potentially harmful combination of experiments and agents |

## Plants

| | |
|---|---|
| Seed stocks | For whole-genome resequencing, 820 inbred lines of cauliflower and B. oleracea relatives were collected, which were developed and stored in the Tianjin Kernel Vegetable Research Institute, Tianjin Academy of Agricultural Sciences, China. |
| Novel plant genotypes | There is no novel plant genotype. |
| Authentication | *Describe any authentication procedures for each seed stock used or novel genotype generated. Describe any experiments used to assess the effect of a mutation and, where applicable, how potential secondary effects (e.g. second site T-DNA insertions, mosiacism, off-target gene editing) were examined.* |

## ChIP-seq

### Data deposition

☐ Confirm that both raw and final processed data have been deposited in a public database such as GEO.

☐ Confirm that you have deposited or provided access to graph files (e.g. BED files) for the called peaks.

| | |
|---|---|
| Data access links<br>*May remain private before publication.* | *For "Initial submission" or "Revised version" documents, provide reviewer access links. For your "Final submission" document, provide a link to the deposited data.* |
| Files in database submission | *Provide a list of all files available in the database submission.* |
| Genome browser session<br>(e.g. UCSC) | *Provide a link to an anonymized genome browser session for "Initial submission" and "Revised version" documents only, to enable peer review. Write "no longer applicable" for "Final submission" documents.* |

### Methodology

| | |
|---|---|
| Replicates | *Describe the experimental replicates, specifying number, type and replicate agreement.* |
| Sequencing depth | *Describe the sequencing depth for each experiment, providing the total number of reads, uniquely mapped reads, length of reads and whether they were paired- or single-end.* |
| Antibodies | *Describe the antibodies used for the ChIP-seq experiments; as applicable, provide supplier name, catalog number, clone name, and lot number.* |
| Peak calling parameters | *Specify the command line program and parameters used for read mapping and peak calling, including the ChIP, control and index files used.* |
| Data quality | *Describe the methods used to ensure data quality in full detail, including how many peaks are at FDR 5% and above 5-fold enrichment.* |
| Software | *Describe the software used to collect and analyze the ChIP-seq data. For custom code that has been deposited into a community repository, provide accession details.* |

# Flow Cytometry

## Plots

Confirm that:

☐ The axis labels state the marker and fluorochrome used (e.g. CD4-FITC).

☐ The axis scales are clearly visible. Include numbers along axes only for bottom left plot of group (a 'group' is an analysis of identical markers).

☐ All plots are contour plots with outliers or pseudocolor plots.

☐ A numerical value for number of cells or percentage (with statistics) is provided.

## Methodology

| | |
|---|---|
| Sample preparation | *Describe the sample preparation, detailing the biological source of the cells and any tissue processing steps used.* |
| Instrument | *Identify the instrument used for data collection, specifying make and model number.* |
| Software | *Describe the software used to collect and analyze the flow cytometry data. For custom code that has been deposited into a community repository, provide accession details.* |
| Cell population abundance | *Describe the abundance of the relevant cell populations within post-sort fractions, providing details on the purity of the samples and how it was determined.* |
| Gating strategy | *Describe the gating strategy used for all relevant experiments, specifying the preliminary FSC/SSC gates of the starting cell population, indicating where boundaries between "positive" and "negative" staining cell populations are defined.* |

☐ Tick this box to confirm that a figure exemplifying the gating strategy is provided in the Supplementary Information.

# Magnetic resonance imaging

## Experimental design

| | |
|---|---|
| Design type | *Indicate task or resting state; event-related or block design.* |
| Design specifications | *Specify the number of blocks, trials or experimental units per session and/or subject, and specify the length of each trial or block (if trials are blocked) and interval between trials.* |
| Behavioral performance measures | *State number and/or type of variables recorded (e.g. correct button press, response time) and what statistics were used to establish that the subjects were performing the task as expected (e.g. mean, range, and/or standard deviation across subjects).* |

## Acquisition

| | |
|---|---|
| Imaging type(s) | *Specify: functional, structural, diffusion, perfusion.* |
| Field strength | *Specify in Tesla* |
| Sequence & imaging parameters | *Specify the pulse sequence type (gradient echo, spin echo, etc.), imaging type (EPI, spiral, etc.), field of view, matrix size, slice thickness, orientation and TE/TR/flip angle.* |
| Area of acquisition | *State whether a whole brain scan was used OR define the area of acquisition, describing how the region was determined.* |

Diffusion MRI    ☐ Used    ☐ Not used

## Preprocessing

| | |
|---|---|
| Preprocessing software | *Provide detail on software version and revision number and on specific parameters (model/functions, brain extraction, segmentation, smoothing kernel size, etc.).* |
| Normalization | *If data were normalized/standardized, describe the approach(es): specify linear or non-linear and define image types used for transformation OR indicate that data were not normalized and explain rationale for lack of normalization.* |
| Normalization template | *Describe the template used for normalization/transformation, specifying subject space or group standardized space (e.g. original Talairach, MNI305, ICBM152) OR indicate that the data were not normalized.* |
| Noise and artifact removal | *Describe your procedure(s) for artifact and structured noise removal, specifying motion parameters, tissue signals and physiological signals (heart rate, respiration).* |

| Volume censoring | *Define your software and/or method and criteria for volume censoring, and state the extent of such censoring.* |
|---|---|

## Statistical modeling & inference

| Model type and settings | *Specify type (mass univariate, multivariate, RSA, predictive, etc.) and describe essential details of the model at the first and second levels (e.g. fixed, random or mixed effects; drift or auto-correlation).* |
|---|---|
| Effect(s) tested | *Define precise effect in terms of the task or stimulus conditions instead of psychological concepts and indicate whether ANOVA or factorial designs were used.* |

Specify type of analysis: ☐ Whole brain  ☐ ROI-based  ☐ Both

| Statistic type for inference | *Specify voxel-wise or cluster-wise and report all relevant parameters for cluster-wise methods.* |
|---|---|

(See Eklund et al. 2016)

| Correction | *Describe the type of correction and how it is obtained for multiple comparisons (e.g. FWE, FDR, permutation or Monte Carlo).* |
|---|---|

## Models & analysis

| n/a | Involved in the study |
|---|---|
| ☐ | ☐ Functional and/or effective connectivity |
| ☐ | ☐ Graph analysis |
| ☐ | ☐ Multivariate modeling or predictive analysis |

| Functional and/or effective connectivity | *Report the measures of dependence used and the model details (e.g. Pearson correlation, partial correlation, mutual information).* |
|---|---|
| Graph analysis | *Report the dependent variable and connectivity measure, specifying weighted graph or binarized graph, subject- or group-level, and the global and/or node summaries used (e.g. clustering coefficient, efficiency, etc.).* |
| Multivariate modeling and predictive analysis | *Specify independent variables, features extraction and dimension reduction, model, training and evaluation metrics.* |

