## [Peer Review File · Nature Genetics]

Peer Review Information

Manuscript Title: Genomic analyses reveal the stepwise domestication and genetic mechanism of curd biogenesis in cauliflower

Corresponding author name(s): Professor Tao Lin, Professor Rui Chen, Professor Deling Sun

Reviewer Comments & Decisions:

Decision Letter, initial version:

25th Apr 2023

Dear Professor Lin,

Your Article, "Genomic analyses reveal the stepwise domestication and genetic mechanism of curd biogenesis in cauliflower" has now been seen by 3 referees. You will see from their comments below that while they find your work of interest, some important points are raised. We are interested in the possibility of publishing your study in Nature Genetics, but would like to consider your response to these concerns in the form of a revised manuscript before we make a final decision on publication.

To guide the scope of the revisions, the editors discuss the referee reports in detail within the team with a view to identifying key priorities that should be addressed in revision. In this case, we think all three referees have provided constructive reviews aimed at strengthening the analyses and improving the presentation. We particularly ask that you improve the novelty and biological insight of the study, and address all referee comments as thoroughly as possible with appropriate revisions. We hope that you will find the prioritized set of referee points to be useful when revising your study. Please do not hesitate to get in touch if you would like to discuss these issues further.

We therefore invite you to revise your manuscript taking into account all reviewer and editor comments. Please highlight all changes in the manuscript text file. At this stage we will need you to upload a copy of the manuscript in MS Word .docx or similar editable format.

*2) If you have not done so already please begin to revise your manuscript so that it conforms to our Article format instructions, available here.
Refer also to any guidelines provided in this letter.

Please be aware of our guidelines on digital image standards.

[redacted]

We hope to receive your revised manuscript within 3 to 6 months. If you cannot send it within this time, please let us know.

Sincerely,
Wei

Wei Li, PhD
Senior Editor
Nature Genetics

New York, NY 10004, USA
www.nature.com/ng

Reviewers' Comments:

Reviewer #1:

Remarks to the Author:

Review of Chen et al., "Genomic analyses reveal the stepwise domestication and genetic mechanism of curd biogenesis in cauliflower.

Overview:

The authors report a new, high-quality genome of the cauliflower lineage of *B. oleracea*. Using large-scale population data, they describe their inferences of the ancestry and population structure of *B. oleracea*, as well as using GWAS techniques to identify genes involved in the developmental pathways underlying the morphological differences between broccoli and cauliflower.

Major comments:

The authors have performed an important service in providing this new, high-quality cauliflower genome. I have a few technical concerns with some aspects of the manuscript, but I find it to be generally sound and using appropriate analysis techniques. However, much of the manuscript reports previously known information on the history of this clade--see especially (Mabry, et al. 2021) without any particular reason to see these new analyses as superior enough to dramatically change our confidence in the conclusions.

Hence, the first part of the manuscript is sound but not particularly novel. The analyses of the genetic and morphological differences between broccoli and cauliflower are of great intrinsic interest, but the analyses here are effectively giving candidate genes for future study—nothing in the results or the analyses performed are sufficient to fully dissect and describe if and how these genes are driving the morphological differences. Hence, it is difficult to be overly excited about the manuscript in its current form.

Minor points:

The manuscript could use further editing for style: for instance lines 198, 259, and 387.

Phylogenetic models are built to be used with full gene sequence data. Applying them to SNP positions where every alignment column has at least two different variants thus does not meet the model assumptions. In fact, the authors approach has been used elsewhere and seems to work well, but it would be wise to note this potential issue in the text.

On lines 356 and following, the authors seem not to know what the definition of an ortholog is: namely a pair of genes with a last common ancestor at a speciation event. When comparing *Brassica* and *Arabidopsis*, ortholog identification can be tricky given the intervening hexaploidy, and in many cases "co-orthologs" are present: multi-copy genes in *Brassica* that are single-copy in *Arabidopsis*. The authors's analyses seem to make no effort to address this complexity and seem to be using ortholog as a synonym for homolog, which is not correct.

Reference:

Mabry ME, Turner-Hissong SD, Gallagher EY, McAlvay AC, An H, Edger PP, Moore JD, Pink DAC, Teakle GR, Stevens CJ, et al. 2021. The Evolutionary History of Wild, Domesticated, and Feral Brassica oleracea (Brassicaceae). *Molecular biology and evolution* 38:4419-4434.

Reviewer #2:

Remarks to the Author:

Deling Sun and his colleagues carried out a typical genomic studies on cauliflower in this study. First they updated their previous draft genome of one cauliflower cultivar to a genome-level assembly. Based on this reference genome, they sequenced a population including 971 accessions and analyze domestication and evolutionary history of cauliflowers. At the end, they further determined some loci/genes associated to seven agronomy phynotypes using a subpopulation of the 971 population by GWAS.

Generally, the manuscript provided an genomic investigationthe with the biggest population in cauliflower up to now and many candidate genes involving the traits of cauliflower. The results are valuable for Brassica vegetatables and other crops.

Major comments:

This is a good and traditional population genomic study and a lot of works have been done. Although contribute a lot to cauliflower crop, no novel or interesting findings for other crops or a more broad field. I failed to find them in its abstract or discussion.

Minor comments:

1. Table 1 is not with much information and can be moved as a supplementary table.
2. Figure 1 is not with much information and can be moved as a supplementary figure or substitute the circle figure to other figures.

Reviewer #3:

Remarks to the Author:

The manuscript by Chen et al provides huge sequencing and resequencing resources, alongside GWAS and haplotype anaylses to evaluate the domestication and trait stabilisation of cauliflower.

It is a massive work and goes far deeper into evaluating this than any previous study and is certainly within the scope of the journal. The data are very strong and clear. The approach is valid, the data high quality in type and presentation, I only wonder if they have provided enough in the way of monogenic validation. However to be honest I am personally convinced by this and feel that the authors are sober in their claims.

The statistics are well carried out.

The conclusions would seem to be fully robust although again would be strengthened with more reverse genetics. This is however the only experiment that I feel is lacking.

One other minor comment. I am not sure the lay reader will know what the curd is. This should be better explained. Similarly, the canonical domestication traits while arguably less interesting may be of more interest to some readers than to the authors (and myself) I feel they should be brought more to the fore in this work.

These textural changes I think are necessary the reverse genetics studies probably only if the authors already have started them, it would be a shame to delay publication overly if they have not.

Author Rebuttal to Initial comments

Responses to Referees

Reviewers' Comments:

Reviewer #1:

Remarks to the Author:

Review of Chen et al., “Genomic analyses reveal the stepwise domestication and genetic mechanism of curd biogenesis in cauliflower”.

Overview:

The authors report a new, high-quality genome of the cauliflower lineage of *B. oleracea*. Using large-scale population data, they describe their inferences of the ancestry and population structure of *B. oleracea*, as well as using GWAS techniques to identify genes involved in the developmental pathways underlying the morphological differences between broccoli and cauliflower.

Response: Many thanks! We hope this study is a good start for the scientists to improve cauliflower breeding and related scientific researches.

Major comments:

The authors have performed an important service in providing this new, high-quality cauliflower genome. I have a few technical concerns with some aspects of the manuscript, but I find it to be generally sound and using appropriate analysis techniques. However, much of the manuscript reports previously known information on the history of this clade--see especially (Mabry, et al. 2021) without any particular reason to see these new analyses as superior enough to dramatically change our confidence in the conclusions.

Response: Thanks for the comments! Our results supported the Mabry’s study about the wild ancestral position of *B. cretica* in *Brassica*. However, we achieved more informative and novel results based on the larger sample size and whole-genome resequencing data, including the landscape of *B. oleracea* phylogenetic relationships, the closest wild ancestral identity of *B. cretica*, the evolution history of Chinese kale, and genome evolution of cauliflower. We have revised the corresponding parts in this manuscript.

In particular, we comprehensively dissected the phylogenetic structure of *B. oleracea* subspecies, especially for cauliflower populations, and provided genomic evidence for the early divergence of Chinese kale in the section “Evolutionary relationships among *B. oleracea* subspecies” and the ancestral identity of *B. cretica* in the section “Genomic evidence for the wild ancestor of *B. oleracea*”. First, we found some discrepancies about biological characterization and phylogenetic clustering of *B. cretica* between our study and the Mabry’s. Our phylogenetic tree showed four *B. cretica* accessions was tightly clustered and reasonable by using the whole genome SNPs and appropriate outgroup accessions. Second, we performed a new strategy of consensus genotype analysis based on the whole genome SNPs. The results showed *B. cretica* might be the closest wild ancestor of all *B. oleracea* subspecies, which is consistent with the Mabry’s deduction. Third, we resequenced more Chinese kale (Clade 1) accessions (59) than other previous studies, and found that Chinese kale is closest to the phylogenetic root and occupies a distinct position in the PCA results for the first time. It is consistent with the historical record that Chinese kale was introduced to China from Europe during the Northern and Southern Dynasties (~AD 420-589) and evolved as an independent population. Last but not least, we investigated 726 cauliflower samples, which were far more than those in the Mabry’s work (18 cauliflower accessions), and performed more detailed individual classification according to plant architecture and maturity levels. In short, we offered novel evidence and valuable information for better understanding the evolutionary relationships among *B. oleracea* subspecies.

Hence, the first part of the manuscript is sound but not particularly novel. The analyses of the genetic and morphological differences between broccoli and cauliflower are of great intrinsic interest, but the analyses here are effectively giving candidate genes for future study—nothing in the results or the analyses performed are sufficient to fully dissect and describe if and how these genes are driving the morphological differences. Hence, it is difficult to be overly excited about the manuscript in its current form.

Response: Thanks for the reviewer’s suggestions. We simplified the first part of the manuscript and moved Figure 1 and Table 1 to supplementary files. To achieve more evidence for these candidate genes potentially controlling curd development, we complemented a batch of

functional experiments and subsequent analyses including two F₂ segregating populations and allele frequency among the populations, as well as knockout and overexpression of candidate genes in the revised manuscript. As a result, we created two F₂ segregating populations (Chinese kale × Cauliflower) and performed bulked segregant analysis (BSA) to further validate eight important loci during Curd-emergence stage and three during Curd-improvement stage (new **Figure 3e,f**). In addition, we overexpressed *FUL2* in cauliflower to determine its potential regulatory function. So far, we didn't find obvious malformation or developmental failures of the curd organ in the early stage of T₀ transgenic plants, suggesting the complex networks underlying curd biogenesis with multiple regulators (data not shown). This issue is a major concern for us, and we will focus on investigating these candidate genes in the next work.

Minor points:

The manuscript could use further editing for style: for instance, lines 198, 259, and 387.

Response: Many thanks for the critical reading. We have corrected them thoroughly in the revised manuscript.

Phylogenetic models are built to be used with full gene sequence data. Applying them to SNP positions where every alignment column has at least two different variants thus does not meet the model assumptions. In fact, the authors' approach has been used elsewhere and seems to work well, but it would be wise to note this potential issue in the text.

Response: We agree with the reviewer's suggestion. To better elucidate this issue, we extracted 4,395 highly identical single-copy genes among four genomes (Chinese kale, TO1000DH3; Cabbage, 02-12; Broccoli, HDEM; and Cauliflower, C-8 V2). Then, we picked up entire SNPs located in these single-copy genes (371,164 SNPs, missing rate < 1%) and isolated full pseudo-CDS sequences of these single-copy genes to construct phylogenetic trees, respectively. As expected, two phylogenetic trees showed a high degree of similarity with our results based on 4d-SNPs data, suggesting that this strategy is accurate and reliable for phylogenetic analysis (**Response Figure 1**). In addition, the approach based on the 4d-SNPs data had been successfully applied in previous studies (Lin et al., 2014; Duan et al., 2017; Zhao et al., 2019). We added the corresponding explanations in the **Methods** section.

Figure 1. The comparisons of the phylogenetic trees constructed using different approaches. a, The phylogenetic tree generated using 69,275 SNPs at fourfold-degenerate sites. **b,** The phylogenetic tree built using 371,164 SNPs located in 4,395 single-copy genes. **c,** The phylogenetic tree built using the full CDS sequences of the 4,395 single-copy genes.

References

Lin, et al. Genomic analyses provide insights into the history of tomato breeding[J]. *Nature Genetics*, 2014, 46(11): 1220-1226.

Duan, et al. Genome re-sequencing reveals the history of apple and supports a two-stage model for fruit enlargement[J]. *Nature communications*, 2017, 8(1): 249.

Zhao, et al. A comprehensive genome variation map of melon identifies multiple domestication events and loci influencing agronomic traits[J]. *Nature Genetics*, 2019, 51(11): 1607-1615.

On lines 356 and following, the authors seem not to know what the definition of an ortholog is: namely a pair of genes with a last common ancestor at a speciation event. When comparing Brassica and Arabidopsis, ortholog identification can be tricky given the intervening hexaploidy, and in many cases “co-orthologs” are present: multi-copy genes in Brassica that are single-copy in Arabidopsis. The authors’s analyses seem to make no effort to address this complexity and seem to be using ortholog as a synonym for homolog, which is not correct.

Response: Thanks for the suggestion. We carefully modified the “ortholog” and “homolog” words for proper description as suggested and highlighted them in the revised manuscript.

Reviewer #2:

Remarks to the Author:

Deling Sun and his colleagues carried out a typical genomic studies on cauliflower in this study. First they updated their previous draft genome of one cauliflower cultivar to a genome-level assembly. Based on this reference genome, they sequenced a population including 971 accessions and analyze domestication and evolutionary history of cauliflowers. At the end, they further determined some loci/genes associated to seven agronomy phenotypes using a subpopulation of the 971 population by GWAS.

Generally, the manuscript provided a genomic investigation the with the biggest population in cauliflower up to now and many candidate genes involving the traits of cauliflower. The results are valuable for Brassica vegetables and other crops.

Response: We thank the reviewer for the comments on our work.

Major comments:

This is a good and traditional population genomic study and a lot of works have been done. Although contribute a lot to cauliflower crop, no novel or interesting findings for other crops or a more broad field. I failed to find them in its abstract or discussion.

Response: Thanks for the reviewer's suggestions. Cauliflower is an interesting crop that have distinctive expanded inflorescences (we called curd), which make it possibly an ideal model plant for studying inflorescences development. In this study, we provided novel and interesting findings on the stepwise domestication (Curd-emergence and Curd-improvement) and detailed classification of cauliflower populations, wild ancestor prediction, and the genetic mechanisms of curd biogenesis and important agronomic traits. Moreover, we identified a zinc finger protein (BOB06G135460) responsible for stem height (SH) and three significantly associated biomass traits. Further functional experiments demonstrated that *BOB06G135460* positively regulated SH in cauliflower (new **Fig. 4i,j** and new **Supplementary Fig. 13**), which makes contribution to biomass breeding in crops. In short, these results could offer important clues for dissecting the genetic mechanisms of florescent development and biomass accumulation in crops. We revised the abstract and discussion sections to emphasize novel and important findings in this manuscript.

Minor comments:

1. Table 1 is not with much information and can be moved as a supplementary table.

Response: Thanks. We revised it as suggested.

2. Figure 1 is not with much information and can be moved as a supplementary figure or substitute the circle figure to other figures.

Response: We revised it as suggested.

Reviewer #3:

Remarks to the Author:

The manuscript by Chen et al provides huge sequencing and resequencing resources, alongside GWAS and haplotype analyses to evaluate the domestication and trait stabilization of cauliflower.

It is a massive work and goes far deeper into evaluating this than any previous study and is certainly within the scope of the journal. The data are very strong and clear. The approach is valid, the data high quality in type and presentation, I only wonder if they have provided enough in the way of monogenic validation. However, to be honest I am personally convinced by this and feel that the authors are sober in their claims.

The statistics are well carried out.

Response: We thank the reviewer for the positive comments on our work. We extended this study to include some statistically robust population-level analysis and a batch of functional experiments. To achieve more evidence for these candidate genes potentially controlling curd development, we created two F₂ segregating populations (Curd-less × White-curd) and performed bulked segregant analysis (BSA) to further validate eight important loci during Curd-emergence stage and three during Curd-improvement stage (new **Figure 3e,f**). In addition, we also demonstrated that *BOB06G135460* positively regulated stem height in cauliflower using overexpression and knockout strategies. As a result, stem height (SH) was significantly suppressed in three knockout lines (new **Fig. 4i,j**). Correspondingly, the transgenic T₀ seedlings displayed an obvious elongated phenotype due solely to the overexpression of *BOB06G135460* (new **Supplementary Fig. 13**). We added the results of monogenic validation in the revised manuscript.

The conclusions would seem to be fully robust although again would be strengthened with more reverse genetics. This is however the only experiment that I feel is lacking.

Response: We agree with the reviewer's suggestions. In the revised manuscript, we complemented several functional experiments including bulked segregant analysis (BSA) using different populations, overexpression and knockout of the candidate genes involving in curd formation and SH trait. Part of the expected outcomes were obtained, and we added these results in the revised manuscript. These supplementary evidences further supported our views and made the conclusion more solid and convinced.

One other minor comment. I am not sure the lay reader will know what the curd is. This should

be better explained. Similarly, the canonical domestication traits while arguably less interesting may be of more interest to some readers than to the authors (and myself) I feel they should be brought more to the fore in this work.

Response: Many thanks. We added explanatory statements to describe the curd organ and each agronomic traits in the revised manuscript to help readers better understand.

These textural changes I think are necessary the reverse genetics studies probably only if the authors already have started them, it would be a shame to delay publication overly if they have not.

Response: Thanks for the comments and critical reading. We have achieved some reverse genetic results that robustly prove our conclusions. We added these experimental evidences in the revised manuscript (new **Figure 3 and 4**).

Decision Letter, first revision:

16th Feb 2024

Dear Dr. Lin,

Thank you for submitting your revised manuscript "Genomic analyses reveal the stepwise domestication and genetic mechanism of curd biogenesis in cauliflower" (NG-A62085R). It has now been seen by the original referees and their comments are below. The reviewers find that the paper has improved in revision, and therefore we'll be happy in principle to publish it in Nature Genetics, pending minor revisions to comply with our editorial and formatting guidelines.

Sincerely,
Wei

Wei Li, PhD
Senior Editor

Nature Genetics
New York, NY 10004, USA
www.nature.com/ng

Reviewer #1 (Remarks to the Author):

Reviewer #2 (Remarks to the Author):

I have read their responses and revised manuscript and I am happy with their current version and no further comments

Reviewer #3 (Remarks to the Author):

The revised manuscript more than satisfactorily addresses my previous concerns. The newly added data and the textural rewrite is fully satisfactory. I have no further comments.

Author Rebuttal, first revision:

Responses to Referees

Reviewers' Comments:

Reviewer #1:

None

Reviewer #2:

I have read their responses and revised manuscript and I am happy with their current version and no further comments

Response: Thanks. Your previous comments are greatly appreciated.

Reviewer #3:

The revised manuscript more than satisfactorily addresses my previous concerns. The newly added data and the textural rewrite is fully satisfactory. I have no further comments.

Response: Many thanks! We hope this study is a good start for the scientists to improve cauliflower breeding and related scientific researches.

Final Decision Letter:

4th Apr 2024

Dear Dr. Lin,

I am delighted to say that your manuscript "Genomic analyses reveal the stepwise domestication and genetic mechanism of curd biogenesis in cauliflower" has been accepted for publication in an upcoming issue of Nature Genetics.

Your paper will be published online after we receive your corrections and will appear in print in the next available issue. You can find out your date of online publication by contacting the Nature Press Office (press@nature.com) after sending your e-proof corrections.

Before your paper is published online, we shall be distributing a press release to news organizations

worldwide, which may very well include details of your work. We are happy for your institution or funding agency to prepare its own press release, but it must mention the embargo date and Nature Genetics. Our Press Office may contact you closer to the time of publication, but if you or your Press Office have any enquiries in the meantime, please contact press@nature.com.

Please note that *Nature Genetics* is a Transformative Journal (TJ). Authors may publish their research with us through the traditional subscription access route or make their paper immediately open access through payment of an article-processing charge (APC). Authors will not be required to make a final decision about access to their article until it has been accepted. Find out more about Transformative Journals

Authors may need to take specific actions to achieve compliance with funder and institutional open access mandates. If your research is supported by a funder that requires immediate open access (e.g. according to Plan S principles) then you should select the gold OA route, and we will direct you to the compliant route where possible. For authors selecting the subscription publication route, the journal's standard licensing terms will need to be accepted, including [a href="https://www.nature.com/nature-portfolio/editorial-policies/self-archiving-and-license-to-publish](https://www.nature.com/nature-portfolio/editorial-policies/self-archiving-and-license-to-publish). Those licensing terms will supersede any other terms that the author or any third party may assert apply to any version of the manuscript.

If you have not already done so, we invite you to upload the step-by-step protocols used in this

manuscript to the Protocols Exchange, part of our on-line web resource, natureprotocols.com. If you complete the upload by the time you receive your manuscript proofs, we can insert links in your article that lead directly to the protocol details. Your protocol will be made freely available upon publication of your paper. By participating in natureprotocols.com, you are enabling researchers to more readily reproduce or adapt the methodology you use. Natureprotocols.com is fully searchable, providing your protocols and paper with increased utility and visibility. Please submit your protocol to <https://protocolexchange.researchsquare.com/>. After entering your nature.com username and password you will need to enter your manuscript number (NG-A62085R1). Further information can be found at <https://www.nature.com/nature-portfolio/editorial-policies/reporting-standards#protocols>

Sincerely,
Wei

Wei Li, PhD
Senior Editor
Nature Genetics
New York, NY 10004, USA
www.nature.com/ng